# Detecting Any Human-Object Interaction Relationship: Universal HOI Detector with Spatial Prompt Learning on Foundation Models

**Yichao Cao**
Southeast University

Qingfei Tang
Nanjing Enbo Tech.

Xiu Su[*]
University of Sydney

Chen Song
Nanjing Enbo Tech.

Shan You
SenseTime

Xiaobo Lu
Southeast University

Chang Xu
University of Sydney

## Abstract

Human-object interaction (HOI) detection aims to comprehend the intricate relationships between humans and objects, predicting $< human, action, object >$ triplets, and serving as the foundation for numerous computer vision tasks. The complexity and diversity of human-object interactions in the real world, however, pose significant challenges for both annotation and recognition, particularly in recognizing interactions within an open world context. This study explores the universal interaction recognition in an open-world setting through the use of Vision-Language (VL) foundation models and large language models (LLMs). The proposed method is dubbed as ***UniHOI***. We conduct a deep analysis of the three hierarchical features inherent in visual HOI detectors and propose a method for high-level relation extraction aimed at VL foundation models, which we call HO prompt-based learning. Our design includes an HO Prompt-guided Decoder (HOPD), facilitates the association of high-level relation representations in the foundation model with various HO pairs within the image. Furthermore, we utilize a LLM (*i.e.* GPT) for interaction interpretation, generating a richer linguistic understanding for complex HOIs. For open-category interaction recognition, our method supports either of two input types: interaction phrase or interpretive sentence. Our efficient architecture design and learning methods effectively unleash the potential of the VL foundation models and LLMs, allowing UniHOI to surpass all existing methods with a substantial margin, under both supervised and zero-shot settings. The code and pre-trained weights are available at: https://github.com/Caoyichao/UniHOI.

## 1   Introduction

Human-object interaction (HOI) detection [15, 7] is a burgeoning field of research in recent years, which stems from object detection and makes higher demands on high-level visual understanding. It plays a crucial role in many vision tasks, e.g., visual question answering, human-centric understanding, image generation, and activity recognition, to name a few representative ones. An excellent HOI detector must accurately localize all the interacting Human-Object (HO) pairs and recognize their interaction, typically represented as an HOI triplet in the format of $< human, action, object >$.

Despite the optimistic progress made by various HOI detectors [27, 12, 18, 49, 14, 29, 8] in recent years, numerous challenges still persist [45]. Naively mapping diverse interactions to one-hot labels is an exceedingly labor-intensive and costly process, which is prone to the loss of semantic information.

---

[*]Corresponding author (xisu5992@uni.sydney.edu.au). First author (caoyichao@seu.edu.cn).

37th Conference on Neural Information Processing Systems (NeurIPS 2023).

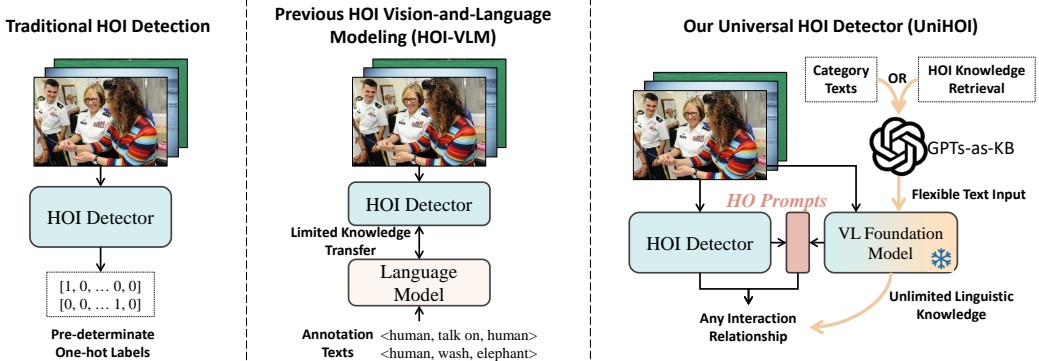

Figure 1: Comparison between HOI detection pipelines. Traditional HOI Detectors rely on manually annotated image datasets for training, which limits their generality and scalability. Previous HOI-VLM methods only achieved limited knowledge transfer from language models to visual HOI detectors, utilizing only a small amount of word embeddings for learning. The proposed UniHOI supports the input of any textual information (annotations or explanations) to detect any interaction relationship, fully unleashing the potential of foundation models and LLMs.

As the number and complexity of interaction categories increase, model optimization becomes increasingly difficult, and the universality of the model is consequently restricted.

Recently, multi-modal learning, especially Vision-and-Language (VL) learning, has become remarkably popular, achieving far-reaching success across a variety of domains [1]. Multi-modal learning is capable of portraying multidimensional information in a more comprehensive manner by describing the same entity or spatio-temporal event through different modalities. Recently, a number of studies have endeavored to employ VL models for the HOI task with the aim of constructing transferable HOI detectors [52, 19, 28, 43, 45, 5]. For instance, PhraseHOI [28] harnessed a pre-trained word embedding model to generate a fixed-length linguistic representation for the HOI task. Unfortunately, these methods harness cross-modal knowledge in an overly constrained manner, falling short of fully exploiting the potential of cross-modal knowledge and LLMs within the HOI detection domain.

Upon rigorous examination, we identify several intrinsic limitations in the existing methodologies: (1) Limited scalability: Model training is overly reliant on annotated data, which restricts them to a finite set of categories. (2) Suboptimal adaptability in zero-shot settings: Even HOI-VLM approaches can only harness a limited number of word embeddings to accommodate unseen categories, thereby curtailing their adaptability. (3) Inability to quickly comprehend complex interactive behaviors from succinct descriptions, as humans can, indicating a dearth of a more universal and flexible framework for HOI detection.

Considering the recent outstanding performance of VL foundation models and LLMs in various fundamental tasks, in this paper, we attempt to harness the capabilities of such large models to facilitate more general HOI detector, as shown in Figure 1. We conduct an in-depth analysis of the clear three-tier visual feature hierarchy in visual HOI detectors, effectively utilizing the foundation models to boost the understanding of high-level interactive semantics within images. The employment of large models bridges the gap between textual and visual modalities, making a knowledge-based universal HOI detector feasible. In summary, the contributions of this paper are as follows:

- We propose the first visual-textual foundation model-based framework for HOI detection (UniHOI), which significantly improves the accuracy and universality compared to previous HOI detectors that solely rely on training from specific datasets. Our UniHOI method is expected to guide the HOI detection field into a new research phase.

- We treat the visual HOI detector as a three-tier visual feature hierarchy and appropriately design an efficient human-object prompt learning method, enabling visual HOI detectors to effectively query interaction tokens associated with specific HO pairs in foundation models, and thus obtain a more universal HO relationship representations.

- We devise a knowledge-based zero-shot HOI recognition method that leverages large models such as GPT to generate descriptive knowledge for specific interaction categories, guiding visual detectors in zero-shot recognition of the corresponding categories. This approach

breaks free from the limitations of traditional zero-shot recognition methods, which can only learn from word embeddings, and offers insights for a more universal open-category HOI recognition system.

## 2  Related Work

**Generic HOI Detection.**  Based on their architectural design, existing HOI (Human-Object Interaction) detection approaches can be broadly categorized into two groups: two-stage methods [49, 7, 12, 13, 27, 18], and one-stage methods [14, 20, 53, 29, 8]. One-stage methods typically adopt a multitask learning strategy to simultaneously execute instance detection and interaction relation modeling [29, 47, 30]. Conversely, two-stage methods initially conduct object detection, followed by interaction relation modeling for all candidate HO pairs. By fully exploiting the capabilities of each module, two-stage methods have showcased superior detection performance [47]. While these methods have significantly propelled the progress of early HOI detection, they still grapple with the issue of limited generality.

**Language Semantics for Vision.**  Driven by the notable success of large-scale models such as GPT-4 [36], the incorporation of language semantics to augment vision models has recently surfaced as a promising avenue in the realm of computer vision tasks [38, 41, 1, 50]. Among these, Vision-and-Language Pre-training (VLP) [34, 24] has evolved into a popular paradigm across numerous vision-and-language tasks, owing to its ability to learn generalizable multimodal representations from extensive image-text data [1, 9, 10]. These methods have recently found applications in multimodal retrieval [11], vision-and-language navigation [2], among others [4]. Nevertheless, implementing effective prompt-based learning on base models, particularly extracting high-order relationships in intricate scenarios, remains crucial for the successful deployment of large models. Thus, investigating how to efficaciously extract specific information from large models is of utmost importance.

**HOI Vision-and-Language Modeling (HOI-VLM).** Despite the moderate success of previous HOI detectors [47, 49, 31], these often perceive interactions as discrete labels, overlooking the rich semantic text information encapsulated in triplet labels.  Of late, a handful of researchers [52, 19, 28, 43, 45, 46] have delved into HOI Vision-and-Language Modeling in an effort to further enhance HOI detection performance. Among these, [52], [19], and [46] have primarily focused on incorporating language prior features into HOI recognition. Meanwhile, RLIP [45] and [43] have proposed the construction of a transferable HOI detector via the VLP approach. Serving as applications and extensions of Vision-and-Language learning within the HOI domain, these HOI-VLM methodologies strive to comprehend the content and relations between visual interaction features and their respective triplet texts. Nevertheless, these methodologies harness cross-modal knowledge in a markedly restricted manner, falling short of fully unleashing the potential of cross-modal knowledge and LLMs within the HOI domain.

## 3  Method

### 3.1  Overview

In traditional HOI detection task, the objective is typically to optimize a detector $\Phi_{\theta_{\mathcal{D}}}$, such as the most commonly-used Transformer-based detectors in recent years. By inputting a query $\mathcal{Q}^{ho}$ related to the Human-Object (HO) pairs, the HOI detector learns the location of the HO pairs and corresponding interactions in images. This optimization objective can be formulated as follows:

$$\min \mathbb{E}_{(\mathcal{I}) \sim \mathcal{X}_{\mathcal{I}}} \left[ \mathcal{L}(\mathcal{GT}, \Phi_{\theta_{\mathcal{D}}}(\mathcal{I}, \mathcal{Q}^{ho})) \right] \tag{1}$$

where $\mathcal{GT}$ and $\mathcal{L}$ are ground-truth labels and overall loss function respectively, $\mathcal{X}_{\mathcal{I}} = \{(\mathcal{I}_i)\}_{i=1}^{|\mathcal{X}|}$ denotes the HOI image dataset, $\mathcal{I}$ represents the input image, and $\theta$ indicates the weights in HOI detector $\Phi_{\theta_{\mathcal{D}}}$. Thus, traditional HOI detectors rely on purely visual detection tasks for optimization.

In this work, our aim is to utilize superior VL foundation models and additional interaction interpretation $\mathcal{T}$ during the training and inference process. In this way, we can improve the optimization objective as follows:

$$\min \mathbb{E}_{(\mathcal{I}, \mathcal{T}) \sim \mathcal{X}} \left[ \mathcal{L}(\mathcal{GT}, \Phi_{\theta_{\mathcal{D}}}(\mathcal{I}, \mathcal{Q}^{ho}), \Phi_{\theta_{\mathcal{F}}}(\mathcal{I}, \mathcal{T}, \mathcal{P}^{ho})) \right] \tag{2}$$

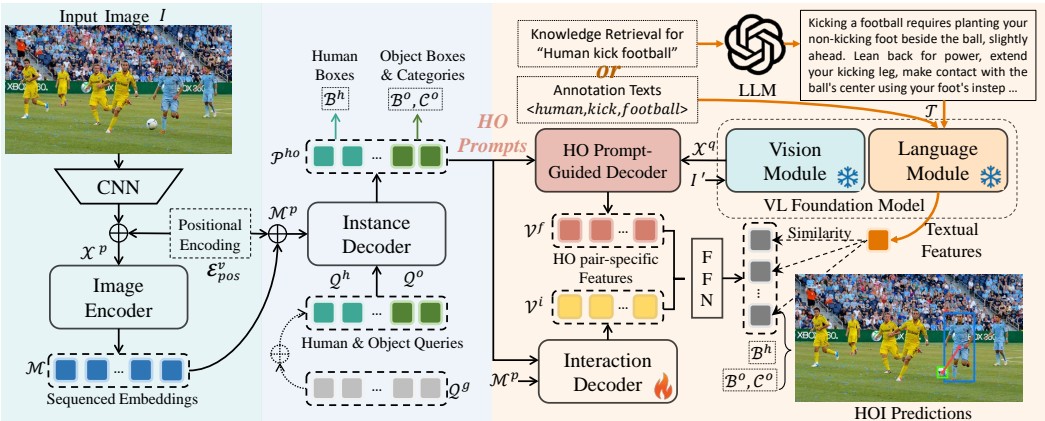

Figure 2: **Overview of the UniHOI.** Its feature hierarchy is divided into three levels. After the first two levels achieve instance detection for humans and objects, we designed a HO spatial prompt learning method, which extracts more generic higher-level relationship representations associated with HO pairs from the VL foundation models. For open-category interaction detection, UniHOI supports annotations or retrieved texts input, and then determine whether the interaction described in the input text occurs between each HO pair through feature similarity measurement.

where $\Phi_{\theta_{\mathcal{F}}}$ denotes the Vision-Language (VL) foundation models and $\mathcal{P}^{ho}$ represents the HOI-specific prompts for $\Phi_{\theta_{\mathcal{F}}}$. $\mathcal{X} = \{(\mathcal{I}_i, \mathcal{T}_i)\}_{i=1}^{|\mathcal{X}|}$ is an image-text corpus, where $\mathcal{I}_i$ denotes the $i$-th input image and $\mathcal{T}_i$ represents the phrase annotations (*e.g.* "Human ride bicycle") in $\mathcal{I}_i$. In zero-shot setting, we also use knowledge retrieval to enrich the texts $\mathcal{T}$. Details will be described in Section 3.4.

In HOI detection, the detector not only needs to accurately locate the position of the interacting HO pair, but also needs to reliably model the interaction relationship occurring in the HO pair. But these two tasks have different requirements on the image feature level: the former focuses more on the edges and contours of instances, while the latter focuses more on high-level relational cues. To present and analyze this complicated process more clearly, we explicitly divide the feature learning in HOI detector into three stages: *basic visual feature extraction*, *instance-level feature learning*, and *high-level relationship modeling*, as shown in Figure 2. The basic visual feature extraction is usually performed by the backbone (*e.g.* CNN). For the instance-level feature learning, we follow previous methodologies and design prompts related to HO locations to carry and transfer the position information, then apply the Hungarian algorithm for matching with the $\mathcal{GT}$, and compute the loss of the bounding box. The third stage, which is the key focus of this study, involves using spatial location prompts related to the HO pairs to conduct targeted prompt learning from the visual model. The details of each component are explained in the following sections.

## 3.2 HO Spatial Prompts Generation

We follow the Transformer-based detection architectures [6] to conduct basic visual feature extraction and instance detection. Fed with an input image $\mathcal{I} \in \mathcal{R}^{H \times W \times C}$, the CNN generates a feature map $\mathcal{X}^v \in \mathcal{R}^{h \times w \times c}$. Then, $\mathcal{X}^v$ is compressed by a projection convolution layer with a kernel size $1 \times 1$. Next, a flatten operator is used to segment patch embeddings $\{x_1^v, x_2^v, \ldots, x_{N^v}^v\}$, where $N^v$ is the number of patch embeddings. The patch embeddings $\{x_1^v, x_2^v, \ldots, x_{N^v}^v\}$ is then linearly projected through a linear transformation $\mathcal{E}^v \in \mathcal{R}^{c \times D^v}$. Thus, the input for instance detection are calculated via summing up the projected embeddings and position embeddings $\mathcal{E}_{pos}^v \in \mathcal{R}^{N^v \times D^v}$:

$$\mathcal{X}^p \in \mathcal{R}^{N^v \times D^v} = [x_1^v \mathcal{E}^v; x_2^v \mathcal{E}^v; \ldots; x_{N^v}^v \mathcal{E}^v] + \mathcal{E}_{pos}^v \tag{3}$$

After patching and sequencing the features extracted by CNN, we follow previous methods to perform self-attention on $\mathcal{X}^p$: $\mathcal{M} = \Phi_{\theta_{\mathcal{IE}}}(\mathcal{X}^p) \in \mathcal{R}^{N^v \times D^v}$. This process is implemented by a Transformer encoder $\Phi_{\theta_{\mathcal{IE}}}$, which is composed of $N$ Transformer encoder layers.

For instance-level feature learning, we deploy an instance decoder $\Phi_{\theta_{\mathcal{ID}}}$ to locate the positions of humans and objects. Based on the $\mathcal{M}$, humans and objects are detected through the human query set

$\mathcal{Q}^h \in \mathcal{R}^{N^q \times D^v}$ and the object query set $\mathcal{Q}^o \in \mathcal{R}^{N^q \times D^v}$ individually. Additionally, a position guided embedding [30] $\mathcal{Q}^g \in \mathcal{R}^{N^q \times D^v}$ is added to assign the human and object queries at the same position as a pair. The calculation process of instance decoder $\Phi_{\theta_{\mathcal{ID}}}$ can be formulated simply as follows:

$$\mathcal{P}^{ho} \in \mathcal{R}^{2N^q \times D^v} = [\mathcal{P}^h, \mathcal{P}^o] = \Phi_{\theta_{\mathcal{ID}}}(\mathcal{M}, [\mathcal{Q}^h + \mathcal{Q}^g, \mathcal{Q}^o + \mathcal{Q}^g]) \tag{4}$$

where the $\mathcal{P}^{ho}$ is the spatial tokens highly correlated with HO pair location information, the $N^q$ is the number of queries. With instance detection head $\mathcal{FFN}s$, we can calculate the human bounding boxs $\mathcal{B}^h \in \mathcal{R}^{N^q \times 4}$ from $\mathcal{P}^h$, object bounding boxs $\mathcal{B}^o \in \mathcal{R}^{N^q \times 4}$ and object categories $\mathcal{C}^o \in \mathcal{R}^{N^q \times N^c}$ from $\mathcal{P}^o$:

$$[\mathcal{B}^h; \mathcal{B}^o; \mathcal{C}^o] = \mathcal{FFN}s([\mathcal{P}^h, \mathcal{P}^o]) \tag{5}$$

where the feature tokens $[\mathcal{P}^h, \mathcal{P}^o]$ are excellent spatial position features for HO pairs. Driven by prompt learning methods, we explore using the HO position features here to perform prompt learning on the VL foundation models.

### 3.3 Prompting Foundation Models for HOI Modeling

Considering the excellent performance of BLIP2 [23] in cross-modal tasks and the fact that its Transformer architecture also integrates well with our HOI framework, in this work we use BLIP2 as the VL foundation model to support the recognition process of HOI. However, accurately extracting HOI related features from VL foundation model is the key to successfully applying VL to HOI tasks.

To reduce the computational cost of the VL foundation model, we first downsample the input image $\mathcal{I}$ to $\mathcal{I}'$ with dimension $W' \times H'$, and then use an image encoder $\Phi_{\theta_{\mathcal{I}}}$ to convert downsampled image into a feature map $\mathcal{X}^f \in \mathcal{R}^{w' \times h' \times c'}$. We then employ Q-Former $\Phi_{\theta_{\mathcal{Q}}}$ in BLIP2 [23] to learn higher-level content $\mathcal{X}^q \in \mathcal{R}^{N^f \times D^f}$ in image:

$$\mathcal{X}^q = \Phi_{\theta_{\mathcal{F}}}(\mathcal{I}') = \Phi_{\theta_{\mathcal{I}}} \circ \Phi_{\theta_{\mathcal{Q}}}(\mathcal{I}, \mathcal{Q}^f) \tag{6}$$

where the $\mathcal{Q}^f$ denotes the query in $\Phi_{\theta_{\mathcal{I}}}$. These modules in $\Phi_{\theta_{\mathcal{I}}}$ are more capable of understanding content in images than regular visiual detectors due to extensive training on a wider range of data.

Then, we consider how to extract the complex interaction representations between specific humans and objects in foundation model. To associate the higher-level information in foundation model with HO pairs, we design a HO Prompt-guided Decoder (HOPD) $\Phi_{\theta_{\mathcal{P}}}$ to correlate the position information of HO pairs with feature tokens in foundation model:

$$\mathcal{V}^f = \Phi_{\theta_{\mathcal{P}}}((\mathcal{P}^h + \mathcal{P}^o)/2, \mathcal{X}^q) \tag{7}$$

where the $\mathcal{V}^f$ denotes the high-level information tokens corresponding to HO pairs. In this manner, the HOI-specific tokens is generated under the guidance of the guidance of the human-object query pairs. This spatial prompts act as query embeddings in HOPD, interact with each other through self-attention layers, and interact with frozen image features through cross-attention layers (inserted every other transformer block in HOPD). In this way, these spatial prompts are fed into HOPD, which in turn goes on to query information about the relationships between their respective corresponding HO pairs. In addition, interaction decoding is performed in the visual HOI detector:

$$\mathcal{V}^i = \Phi_{\theta_{\mathcal{IN}}}((\mathcal{P}^h + \mathcal{P}^o)/2, \mathcal{M}) \tag{8}$$

Finally, we concatenate $\mathcal{V}^f$ and $\mathcal{V}^f$ as the visual representations of the HO pairs and feed them to a FFN to predict the interaction category. For model training, we adhere to the precedent set by query-based methods [6], utilizing the Hungarian algorithm to assign a corresponding prediction for each ground-truth instance. The loss and hyperparameters used for model optimization in this paper follow those of previous work [30]. More details are provided in the Appendix.

### 3.4 Knowledge Retrieval for HOI Reasoning in Open World

In previous zero-shot HOI detectors, the recognition of unseen categories is usually achieved by directly using word embeddings and visual features for similarity measures. The problem with this approach is that it relies only on simple word embeddings, which may not be sufficiently detailed

and rich in the representation of HOIs. The model may not recognize some special categories well enough. Inspired by the human understanding of interactions, we propose a method for HOI detection in an open world based on descriptive text.

The impressive performance of large-scale pretrained language models, as well as the potentially enormous amount of implicitly stored knowledge, raises extensive attention about using language models as an alternative to conventional structured knowledge bases (LMs-as-KBs). In particular, the rapid development of the chatGPT family of LLM models in the last two years inspired us to obtain richer and more comprehensive knowledge descriptions, called GPTs-as-KBs. Theoretically, the performance of the visual model can be improved theoretically by transferring a larger amount of knowledge that can be effectively integrated into the visual model. Given an input image $\mathcal{I}^i$ and corresponding descriptions $\mathcal{T}_i$ in $\mathcal{X} = \{(\mathcal{I}_i, \mathcal{T}_i)\}_{i=1}^{|\mathcal{X}|}$, different phrase descriptions are linked with different HO pairs. We propose expanding the existing phrase descriptions $\mathcal{T}^i$ to obtain richer and more comprehensive knowledge descriptions $\mathcal{K}^i$. These descriptive knowledge facilitate more accurate interaction descriptions in zero-shot and open-vocabulary HOI recognition.

To obtain richer and more comprehensive knowledge descriptions, we design a knowledge retrieval process for LLM. We have designed a knowledge query statement for chatGPT, such as "`Knowledge retrieve for` $verb\_objection$`, limited to` $N$ `words`", to guide LLMs in explaining the interaction categories we are interested in. In this way, the transferable textual knowledge will be richer and the interpretation of the interactions will be closer to the way humans understand them. As an illustration, let us consider the phrase "Human ride bicycle" and "Human throw frisbee" . The knowledge retrieval process yields the following example result:

- ***Riding a bicycle*** *involves balance, coordination, and physical exertion. The rider mounts the bike, propels forward by pushing pedals with their feet. Steering is achieved by turning handlebars. Brakes slow or stop the bike. Helmets are worn for safety.*

- ***Throwing a frisbee*** *involves grasping the disc, typically with a forehand grip, then swinging the arm and releasing the frisbee at the right moment for it to glide through the air. Direction and distance depend on the angle and speed of the throw. It's a common recreational activity.*

These detailed text descriptions can better guide the learning of visual features in open-category recognition. A more detailed open-category inference process is provided in the Appendix.

## 4 Experiments

### 4.1 Implementation Details

We evaluate our model on two public benchmarks, HICO-Det [7] and V-COCO [15]. We adopt the BLIP2 [23] model as the VL foundation model in our UniHOI. We map the 768-dimensional features output by Q-Former in BLIP2 to 256 dimensions to adapt to the feature dimension of the visual HOI detector. The number of layers in the HO Prompt-guided Decoder is identical to that in the Interaction Decoder. The output features of these two decoders are concatenated to generate the final category prediction. The model's optimization method, learning rate, training epochs, weight decay, and loss weights, among other hyperparameters, are set in accordance with previous methods to ensure fair comparison. Aside from the ablation studies in Section 4.6 where we employed knowledge retrieval to further enhance model performance, in all other experiments we continue to use annotated phrases for model learning and inference to ensure fair comparison with other methods. All the experiments conducted with a batchsize of 16 on 8 Tesla V100 GPUs. The details of datasets, evaluation metrics, model parameters, computational overhead, *etc.* will be introduced in Appendix.

### 4.2 HOI Detection in the Closed World

We compare the performance of our UniHOI with the previous representative and state-of-the-art methods such as ViPLO [37], GEN-VLKT [30], and HOICLIP [35]. The detailed results of HICO-DET [7] and V-COCO [15] are summarized in Table 1 and 3, respectively. Generally, there are several observations drawn from different aspects: (*i*) The overall performance of our method is significantly superior to the previous state-of-the-art methods. (*ii*) Whether the visual feature extractor adopts the ResNet-50 or ResNet-101, our method yields excellent performance, especially a stronger backbone

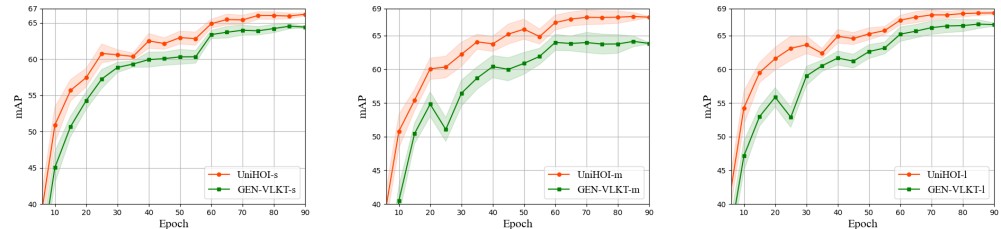

Figure 3: The training process of **UniHOI** and previous State-of-the-Art **GEN-VLKT** [30] on V-COCO dataset.

Table 1: Experimental results on HICO-DET [7]. GEN-VLKT [30] is regarded as baseline.

| Method | Backbone | Default Setting | | | Known Objects Setting | | |
|---|---|---|---|---|---|---|---|
| | | Full | Rare | Non-rare | Full | Rare | Non-rare |
| Two-stage Methods: | | | | | | | |
| ATL [17] | ResNet-50 | 23.81 | 17.43 | 27.42 | 27.38 | 22.09 | 28.96 |
| VSGNet [42] | ResNet-152 | 19.80 | 16.05 | 20.91 | - | - | - |
| DJ-RN [25] | ResNet-50 | 21.34 | 18.53 | 22.18 | 23.69 | 20.64 | 24.60 |
| VCL [16] | ResNet-50 | 23.63 | 17.21 | 25.55 | 25.98 | 19.12 | 28.03 |
| DRG [12] | ResNet-50-FPN | 24.53 | 19.47 | 26.04 | 27.98 | 23.11 | 29.43 |
| IDN [26] | ResNet-50 | 24.58 | 20.33 | 25.86 | 27.89 | 23.64 | 29.16 |
| FCL [18] | ResNet-50 | 25.27 | 20.57 | 26.67 | 27.71 | 22.34 | 28.93 |
| SCG [48] | ResNet-50-FPN | 29.26 | 24.61 | 30.65 | 32.87 | 27.89 | 34.35 |
| UPT [49] | ResNet-50 | 31.66 | 25.90 | 33.36 | 35.05 | 29.27 | 36.77 |
| UPT [49] | ResNet-101 | 32.31 | 28.55 | 33.44 | 35.65 | 31.60 | 36.86 |
| ViPLO-s [37] | ViT-B/32 | 34.95 | 33.83 | 35.28 | 38.15 | 36.77 | 38.56 |
| ViPLO-l [37] | ViT-B/16 | 37.22 | 35.45 | 37.75 | 40.61 | 38.82 | 41.15 |
| One-stage Methods: | | | | | | | |
| PPDM [29] | Hourglass-104 | 21.94 | 13.97 | 24.32 | 24.81 | 17.09 | 27.12 |
| HOI-Trans [54] | ResNet-101 | 26.61 | 19.15 | 28.84 | 29.13 | 20.98 | 31.57 |
| AS-Net [8] | ResNet-50 | 28.87 | 24.25 | 30.25 | 31.74 | 27.07 | 33.14 |
| QPIC [40] | ResNet-101 | 29.90 | 23.92 | 31.69 | 32.38 | 26.06 | 34.27 |
| SSRT [19] | ResNet-101 | 31.34 | 24.31 | 33.32 | - | - | - |
| CDN-S [47] | ResNet-50 | 31.44 | 27.39 | 32.64 | 34.09 | 29.63 | 35.42 |
| CDN-L [47] | ResNet-101 | 32.07 | 27.19 | 33.53 | 34.79 | 29.48 | 36.38 |
| Liu *et al.* [31] | ResNet-50 | 33.51 | 30.30 | 34.46 | 36.28 | 33.16 | 37.21 |
| MUREN [22] | ResNet-50 | 32.87 | 28.67 | 34.12 | 35.52 | 30.88 | 36.91 |
| GEN-VLKT-s [30] | ResNet-50 | 33.75 | 29.25 | 35.10 | 36.78 | 32.75 | 37.99 |
| GEN-VLKT-m [30] | ResNet-101 | 34.78 | 31.50 | 35.77 | 38.07 | 34.94 | 39.01 |
| GEN-VLKT-l [30] | ResNet-101 | 34.96 | 31.18 | 36.08 | 38.22 | 34.36 | 39.37 |
| HOICLIP [35] | ResNet-101 | 34.69 | 31.12 | 35.74 | 37.61 | 34.47 | 38.54 |
| Xie *et al.* (large) [44] | ResNet-101 | 36.03 | 33.16 | 36.89 | 38.82 | 35.51 | 39.81 |
| ***UniHOI-s*** (w/ BLIP2) | ResNet-50 | 40.06 (+6.31) | 39.91 (+10.66) | 40.11 (+5.01) | 42.20 (+5.42) | 42.60 (+9.85) | 42.08 (+4.09) |
| ***UniHOI-m*** (w/ BLIP2) | ResNet-101 | 40.74 (+5.96) | 40.03 (+8.53) | 40.95 (+5.18) | 42.96 (+4.89) | 42.86 (+7.92) | 42.98 (+3.97) |
| ***UniHOI-l*** (w/ BLIP2) | ResNet-101 | 40.95 (+5.99) | 40.27 (+9.09) | 41.32 (+5.24) | 43.26 (+5.04) | 43.12 (+8.76) | 43.25 (+3.88) |

that can bring further improvements. (*iii*) As shown in Tables 2, our approach also has a significant performance lead compared to those methods that rely on additional data sets (*i.e.*, Human Pose [32], and Vision-and-Language [45]). Figure 3 shows the training process of UniHOI-s and GEN-VLKT-s on the V-COCO dataset. Figure 4 shows some prediction cases of UniHOI on HICO-DET.

## 4.3 Comparisons with Methods that Utilize Extra Information

Table 2 presents several methods that utilize extra information (*i.e.*, pose, and language) from additional datasets for HOI detection. Theoretically, the introduction of extra information can indeed further enhance visual models in specific domains. The results demonstrate that the proposed UniHOI still significantly outperforms previous methods.

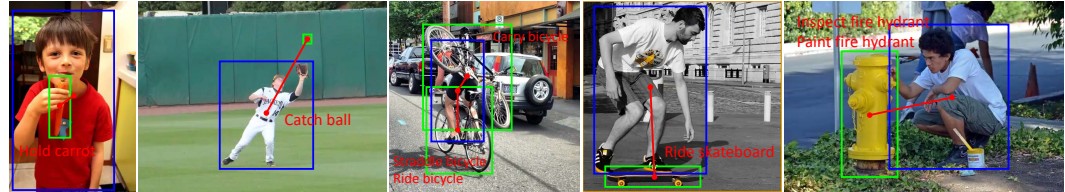

Figure 4: Detection results of our UniHOI in HICO-DET [7].

Table 2: Comparison results with the methods using extra datasets on HICO-DET dataset [7]. For extra datasets, "P" indicates human pose and "L" indicates linguistic knowledge.

| Method | Extras | HICO-DET | | | | | | V-COCO | |
| | | Default Setting | | | Known Objects Setting | | | $AP_{role}^{\#1}$ | $AP_{role}^{\#2}$ |
| | | Full | Rare | Non-rare | Full | Rare | Non-rare | | |
|---|---|---|---|---|---|---|---|---|---|
| FCMNet [32] | P+L | 20.41 | 17.34 | 21.56 | 22.04 | 18.97 | 23.12 | 53.1 | - |
| PD-Net [52] | L | 20.76 | 15.68 | 22.28 | 25.59 | 19.93 | 27.28 | - | - |
| DRG [12] | P | 24.53 | 19.47 | 26.04 | 27.98 | 23.11 | 29.43 | 51.0 | - |
| ConsNet-F [33] | P | 24.39 | 17.10 | 26.56 | - | - | - | 53.2 | - |
| RLIP-ParSeD [45] | L | 30.70 | 24.67 | 32.50 | - | - | - | - | - |
| RLIP-ParSe [45] | L | 32.84 | 26.85 | 34.63 | - | - | - | 61.9 | 64.2 |
| PhraseHOI [28] | L | 30.03 | 23.48 | 31.99 | 33.74 | 27.35 | 35.64 | - | - |
| OCN (large) [46] | L | 31.43 | 25.80 | 33.11 | - | - | - | - | - |
| HOICLIP [35] | L | 34.69 | 31.12 | 35.74 | 37.61 | 34.47 | 38.54 | 63.5 | 64.8 |
| *UniHOI-s* (w/ BLIP2) | L | 40.06 | 39.91 | 40.11 | 42.20 | 42.60 | 42.08 | 65.58 | 68.27 |
| *UniHOI-m* (w/ BLIP2) | L | 40.74 | 40.03 | 40.95 | 42.96 | 42.86 | 42.98 | 67.95 | 70.61 |
| *UniHOI-l* (w/ BLIP2) | L | 40.95 | 40.27 | 41.32 | 43.26 | 43.12 | 43.25 | 68.05 | 70.82 |

## 4.4 Effectiveness for Zero-Shot HOI Detection

**Unseen Composition (UC).** As shown in Table 4, we follow the [30] to conduct three kind of zero-shot experiments on HICO-DET. We first evaluate UniHOI-s with UC setting. Compared with GEN-VLKT [30], our UniHOI-s substantially outperforms previous state-of-the-art on both RF-UC and NF-UC. Among them, UniHOI are 7.32mAP higher than GEN-VLKT on Unseen of RF-UC, and 9.25mAP higher than GEN-VLKT on Seen of NF-UC. This improvement is mainly due to our efficient prompt learning method for the foundation model, so UniHOI can cope well with some unseen categories. In fact, GEN-VLKT also performs some knowledge transfer from CLIP, but it lags behind UniHOI in the effectiveness of knowledge transfer.

**Unseen Object (UO).** For UO setting, compared with GEN-VLKT, UniHOI improves Unseen, Seen, and Full by 9.21, 5.84, and 5.93, respectively. This result proves that our method can still understand the interaction between human and novel objects well.

**Unseen Verb (UV).** On the challenging UV setting, compared to GEN-VLKT, UniHOI improves by 9.21, 5.84 and 5.93 on Unseen, Seen, and Full, respectively. In fact, on the three zero-shot settings mentioned above, the HOI recognition capability of UniHOI have almost surpassed some supervised

Table 3: Experimental results on V-COCO.

| Method | $AP_{role}^{\#1}$ | $AP_{role}^{\#2}$ |
|---|---|---|
| VSGNet [42] | 51.8 | 57.0 |
| IDN [26] | 53.3 | 60.3 |
| UPT [49] | 60.7 | 66.2 |
| VIPLO-s [37] | 60.9 | 66.6 |
| VIPLO-l [37] | 62.2 | 68.0 |
| HOTR [21] | 55.2 | 64.4 |
| QPIC [40] | 58.8 | 61.0 |
| CDN [47] | 63.91 | 65.89 |
| Liu *et al.* [31] | 63.0 | 65.2 |
| GEN-VLKT-s [30] | 62.41 | 64.46 |
| GEN-VLKT-m [30] | 63.28 | 65.58 |
| GEN-VLKT-l [30] | 63.58 | 65.93 |
| HOICLIP [35] | 63.50 | 64.80 |
| Xie *et al.* (large) [44] | 66.50 | 69.90 |
| *UniHOI-s* (w/ BLIP2) | 65.58 (+3.17) | 68.27 (+3.81) |
| *UniHOI-m* (w/ BLIP2) | 67.95 (+4.67) | 70.61 (+5.03) |
| *UniHOI-l* (w/ BLIP2) | 68.05 (+4.47) | 70.82 (+4.89) |

Table 4: Zero-shot results on HICO-DET [7].

| Method | Type | Unseen | Seen | Full |
|---|---|---|---|---|
| Shen *et al.* [39] | UC | 5.62 | - | 6.26 |
| FG [3] | UC | 10.93 | 12.60 | 12.26 |
| ATL [17] | UC | 16.99 | 20.51 | 19.81 |
| VCL [16] | RF-UC | 10.06 | 24.28 | 21.43 |
| ATL [17] | RF-UC | 9.18 | 24.67 | 21.57 |
| FCL [18] | RF-UC | 13.16 | 24.23 | 22.01 |
| GEN-VLKT [30] | RF-UC | 21.36 | 32.91 | 30.56 |
| *UniHOI-s* (w/ BLIP2) | RF-UC | 28.68 (+7.32) | 33.16 (+0.25) | 32.27 (+1.71) |
| VCL [16] | NF-UC | 16.22 | 18.52 | 18.06 |
| ATL [17] | NF-UC | 18.25 | 18.78 | 18.67 |
| FCL [18] | NF-UC | 18.66 | 19.55 | 19.37 |
| GEN-VLKT [30] | NF-UC | 25.05 | 23.38 | 23.71 |
| *UniHOI-s* (w/ BLIP2) | NF-UC | 28.45 (+3.4) | 32.63 (+9.25) | 31.79 (+8.08) |
| GEN-VLKT [30] | UO | 10.51 | 28.92 | 25.63 |
| *UniHOI-s* (w/ BLIP2) | UO | 19.72 (+9.21) | 34.76 (+5.84) | 31.56 (+5.93) |
| GEN-VLKT [30] | UV | 20.96 | 30.23 | 28.74 |
| *UniHOI-s* (w/ BLIP2) | UV | 26.05 (+5.09) | 36.78 (+6.55) | 34.68 (+5.94) |

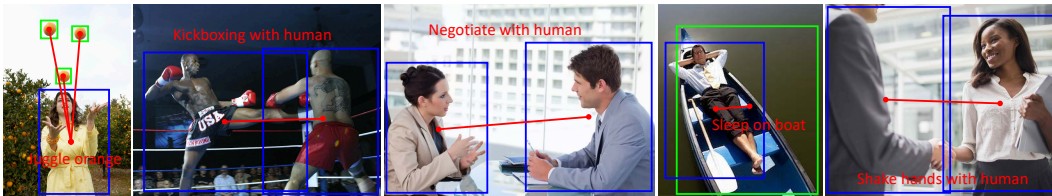

- **Juggling oranges** involves tossing and catching multiple oranges in a specific pattern. Typically, the juggler keeps more oranges in the air than they have hands. It requires good hand-eye coordination, rhythm, and timing. This activity can be entertaining and challenging.
- **Kickboxing** is a hybrid combat sport that combines elements of punching from boxing and kicking from karate or muay thai. Humans participating in kickboxing engage in rigorous physical training, improving their strength, flexibility, and coordination. It's both a competitive sport and fitness regimen.
- **Negotiating with a human** involves communication and compromise to reach a mutual agreement. It requires skills like active listening, articulation, patience, and persuasion. Negotiations can occur in various contexts, including business transactions, conflict resolution, and interpersonal relationships.
- **Sleeping on a boat** can be a unique experience, influenced by factors like the boat's size, stability, and location. The gentle rocking can be soothing, but conditions can vary. It's important to ensure safety precautions, such as wearing lifejackets, are in place.
- **Shaking hands with a human** is a common form of greeting, agreement, or parting. The typical handshake involves a firm grip and brief up and down movement. It's a universal sign of goodwill, respect, and mutual understanding between individuals.

Figure 5: HOI Detection in the Wild. UniHOI is capable of conducting open-category detection based on triplet phrases and also supports the input of descriptive texts retrieved by GPT [36].

Table 5: Ablation Studies on V-COCO [15].

| Method | $AP_{role}^{\#1}$ | $AP_{role}^{\#2}$ |
|---|---|---|
| baseline | 62.41 | 64.46 |
| + VL Foundation Model | 62.91 | 64.83 |
| + HOPD | 65.58 | 68.27 |
| + Knowledge Retrieval | **66.74** | **69.31** |

Table 6: Ablation Studies on HICO-DET (UV).

| Method | Unseen | Seen | Full |
|---|---|---|---|
| baseline | 20.96 | 30.23 | 28.74 |
| + VL Foundation Model | 21.57 | 31.62 | 29.85 |
| + HOPD | 26.05 | 36.78 | 34.68 |
| + Knowledge Retrieval | **27.41** | **37.82** | **35.89** |

methods in Table 1. This once again attests to the superiority of our UniHOI for open-category HOI detection and its powerful capability to understand interactions.

## 4.5 HOI Detection in the Wild

In addition to the zero-shot experiments on the HICO dataset, we also conducted open-category testing on open-world. Our UniHOI method supports flexible text input for detecting corresponding interactions. UniHOI can not only recognize category texts directly (e.g., "Juggle orange") but also take a descriptive text input (which could be human-written or knowledge retrieved from GPT). Consequently, interactions in images are identified through text representation. We present these detection results, along with some GPT-generated description texts, in Figure 5. Our test results indicate that open-vocabulary recognition based on interaction description texts is more reliable, and this phenomenon is also in line with our basic intuition. In the future, we will release our pretrained model and code to enable more extensive testing in the open world.

## 4.6 Ablation Studies

**Effect of Foundation Model.** We conduct ablation studies on the new components within UniHOI (as shown in Tables 5 and 6). The GEN-VLKT-s is regarded as the baseline, and then incorporated various components for testing on the fully supervised V-COCO dataset, as well as the HICO-DET dataset under a zero-shot setting. Initially, we simply integrated the BLIP2 into our method in a sequential manner. We then used learnable queries to extract interaction features from Q-Former and concatenate them with $\mathcal{V}^i$. However, the results did not yield a significant performance improvement.

**Impact of HO Spatial Prompting.** After realizing the alignment issue with the features of HO-pairs in the previous methods, we introduced the HO Prompt-guided Decoder, as presented in Section 3.3. The results clearly illustrate that the introduction of this component significantly improved all performance metrics, thereby validating the effectiveness of our prompting learning strategy.

**Effect of Knowledge Retrieval.** To substantiate the importance of utilizing LLM for knowledge retrieval in creating a more universal HOI detector, we also conduct supplementary experiments in this strategy. The results showed that the UniHOI model, when applying knowledge retrieval, achieved further performance improvements. We believe this approach opens new avenues for enhancement in the field of zero-shot HOI detection.

Table 7: The results of UniHOI equipped with different foundation models (*i.e.,* CLIP [38] and BLIP2 [23]) on V-COCO [15].

| Method | $AP^{\#1}_{role}$ | $AP^{\#2}_{role}$ |
|---|---|---|
| GEN-VLKT-s [30] | 62.41 | 64.46 |
| GEN-VLKT-m [30] | 63.28 | 65.58 |
| GEN-VLKT-l [30] | 63.58 | 65.93 |
| *UniHOI-s* (w/ CLIP) | 62.92 (+0.51) | 65.67 (+1.21) |
| *UniHOI-m* (w/ CLIP) | 64.85 (+1.57) | 67.62 (+2.04) |
| *UniHOI-l* (w/ CLIP) | 64.93 (+1.35) | 67.86 (+1.93) |
| *UniHOI-s* (w/ BLIP2) | **65.58** (+3.17) | **68.27** (+3.81) |
| *UniHOI-m* (w/ BLIP2) | **67.95** (+4.67) | **70.61** (+5.03) |
| *UniHOI-l* (w/ BLIP2) | **68.05** (+4.47) | **70.82** (+4.89) |

**Effect of Different Foundation Model.** To assess the impact of different Foundation Models on the performance of UniHOI, we also provide a comparative analysis between UniHOI equipped with CLIP and BLIP2. For CLIP, we feed its visual output representation into HOPD. However, given that CLIP's cross-modal contrastive training strategy leans more towards image-level representations [38], its output dimensionality is considerably lower than that of BLIP2, resulting in a less detailed feature representation. Experimental results indicate that both the UniHOI versions with CLIP and BLIP2 as the foundational models showed performance enhancements, with the latter exhibiting a more pronounced improvement. These findings underscore the efficacy and generalization of our approach.

## 5 Conclusion

In this work, we propose UniHOI, a universal HOI detection framework. By utilizing large foundation model, spatially prompting the large model, and generalizing the HOI detector to the open world. UniHOI supports open-category interaction relationship recognition from annotation texts or descriptions, which can be flexibly extended to applications in the open world. Extensive experiments on public datasets and diverse settings demonstrate its strong universality - it behaves the strongest open category interaction recognition ability so far. We believe our research will stimulate following research along the universal high-level relationship modeling direction in the future.

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
