The supplementary materials are organized as follows. In Appendix A, we present more detailed descriptions of our UniHOI architecture. In Appendix B, we discuss the motivations for our UniHOI. In Appendix C, we provide an in-depth explanation of differences between our UniHOI and previous CLIP-based methods. In Appendix D, we examine the effects of VL foundation models of different scales. In Appendix E, we provide an detailed explanation of the training and hyperparameter setting. In Appendix F, we present detailed description of the open-category inference. In Appendix G, we present the computational and inference costs of different components in UniHOI. In Appendix H, we outline the datasets and evaluation metrics used in our experiments.

## A    Architecture Details

In this study, the BLIP-2 ViT-L $OPT_{2.7B}$ [23] is adopted as the default Vision-Language (VL) foundation model for all experiments. Specifically, a pre-trained ViT-L/14 is utilized as the image encoder $\Phi_{\theta_{\mathcal{I}}}$ in BLIP2, while the pre-trained OPT [51] serves as the Language Language Model (LLM). An input image, downsampled to $224 \times 224 \times 3$, is fed into the ViT-L, transforming it into $257 \times 1408$ dimensional Image tokens $\mathcal{X}^f$. Subsequently, a lightweight Q-Former $\Phi_{\theta_{\mathcal{Q}}}$ leverages a set of learnable query vectors to extract visual features from $\mathcal{X}^f$. In BLIP2 [23], the Q-Former acts as the bridge between the visual and language modalities, thereby effectively extracting representations that align with the textual context, which in turn facilitates the possibility of open-set HOI detection. In UniHOI, the $32 \times 768$ dimensional output of the Q-Former is passed through a Fully Connected (FC) layer to map it to $32 \times 256$ dimensions, thereby aligning with the feature dimensions of the visual HOI detector.

Following the setup of GEN-VLKT [30], we have implemented three versions of UniHOI: the small version (UniHOI-s), the middle version (UniHOI-m), and the large version (UniHOI-l). Specifically, UniHOI-s employs ResNet-50 as the backbone network, while the latter two, UniHOI-m and UniHOI-l, utilize ResNet-101 as their backbone. For both UniHOI-s and UniHOI-m, the number of layers in the three Decoders—Instance, Interaction, and HO (Human-Object) Prompt-guided Decoder—is set to 3. In contrast, for UniHOI-l, each of the three Decoders comprises 6 decoder layers.

## B    Motivations for Our UniHOI Framework

1) HOI detectors solely relying on supervised learning are inherently constrained in terms of the variety of categories they can handle, thereby limiting their generalizability and scalability.

2) While there have been text-driven zero-shot HOI detectors, they have not adequately addressed the cross-modal alignment issue in an open-world context. The underlying reasons are twofold: First, many methods are trained on a small amount of annotated HOI text, such as PhraseHOI [28], which limits the potential to extend HOI detectors to broader scenarios (see "***Lack of open world knowledge***" in Figure 6). Second, these methods have not achieved HO pair-level cross-modal feature learning, predominantly seen in some CLIP-based methods [38]. This is due to CLIP's cross-modal contrastive learning approach that favors global representations over HO-pair level feature representation (see "***Feature entanglement problem***" in Figure 6). We will elaborate on these details in the next section.

3) The paradigm of utilizing large VL foundation models to drive task-specific operations may become mainstream in the near future. This is because large VL foundation models encapsulate a more comprehensive world knowledge. If they are effectively applied to specific tasks, they can certainly propel advancements in more specialized fields. In the context of HOI detection, a crucial question is how to extract fine-grained higher-level human-object relationship information. How to effectively extract relationship features at the HO pair level from a VL foundation model represents a core issue that large model-driven HOI detection needs to address. In addition, richer descriptive knowledge could theoretically facilitate the understanding of interactions by visual HOI detectors (see "***Flexible knowledge retrieval***" in Figure 6).

In summary, the aforementioned considerations collectively motivated our design for the UniHOI structure. The final experiments have confirmed the validity of these perspectives.

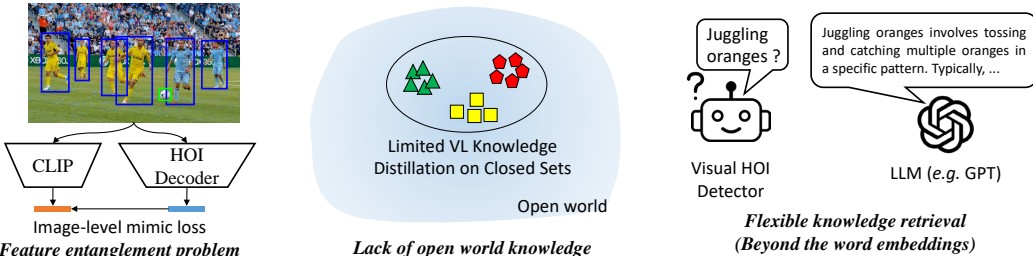

Figure 6: Our UniHOI focuses on the aforementioned issues in the current HOI detection field.

Table 8: Results on V-COCO with the UniHOI-s using different VL Foundation models.

| Methods | $AP_{role}^{\#1}$ | $AP_{role}^{\#2}$ |
|---|---|---|
| GEN-VLKT-s [30] | 62.41 | 64.46 |
| GEN-VLKT-m [30] | 63.28 | 65.58 |
| GEN-VLKT-l [30] | 63.58 | 65.93 |
| *UniHOI-s* (w/ BLIP-2 ViT-L OPT$_{2.7B}$) | 65.58 | 68.27 |
| *UniHOI-s* (w/ BLIP-2 ViT-G OPT$_{6.7B}$) | **65.92** | 68.56 |
| *UniHOI-s* (w/ BLIP-2 ViT-L FlanT5$_{XL}$) | 65.37 | 68.15 |
| *UniHOI-s* (w/ BLIP-2 ViT-G FlanT5$_{XXL}$) | 65.80 | **68.57** |

## C Differences from the Previous CLIP-based Methods

1) Previous CLIP-based methods exhibit limitations in the effectiveness of cross-modal learning. For instance, GEN-VLKT [30] performs visual mimic learning on the entire input image, which directly pools the features of the Interaction Decoder, using the mimic loss to learn the feature distribution of the CLIP Image Encoder. However, an input image may contain numerous human-object (HO) pairs, both interacting and non-interacting, leading to entangled features (see "***Feature entanglement problem***" in Figure 6). Such a method of HOI knowledge distillation is inherently inefficient. Given CLIP's contrastive training strategy, it excels as a model adept at global representation. Therefore, our UniHOI attempts to exploit the knowledge of the VL foundation model from a more meticulous perspective of HO prompt learning.

2) CLIP-based methods' cross-modal learning didn't address the feature alignment problem of HO pairs. If the features at the HO level cannot be aligned, incorporating features from large models directly might lead to limited improvement. For example, HOICLIP [35] tried to leverage CLIP's visual features and learn through cross-attention. However, its improvement compared to GEN-VLKT is still limited.

3) Many methodologies resort to data transfer on limited HOI datasets. For instance, PhraseHOI [28] proposed an excellent idea for Joint Visual-and-Text Modeling. Nevertheless, HOI datasets can't cover the open world, restricting their adaptability to open categories.

4) Another common limitation of previous CLIP-based methods is the reliance on a naive text template. For instance, GEN-VLKT [30] and PhraseHOI [28] both employ the template "`A photo of a person [ACT] [OBJ]`". In this case, the text knowledge that the model can leverage merely comprises two word embeddings, which in turn limits the effectiveness of knowledge transfer. Our UniHOI method can learn more intricate interaction relationships from more diverse interpretive statements (see "***Flexible knowledge retrieval***" in Figure 6).

## D Effects of VL Foundation Models with Different Scales

We build UniHOI variants equipped with other VL foundation models and conduct comparison experiments on the V-COCO [15] dataset to explore the effect of different large models. In Table 8, we show the results of UniHOI-s equipped with different large models. These results indicate that different large models impact HOI recognition capability; generally, all these models promote our UniHOI framework to obtain state-of-the-art results on the V-COCO test set.

## E  Training details and hyperparamters

For model training, we follow the previous query-based methods [6], employing the Hungarian algorithm to assign a matching prediction for each ground-truth. The match incorporates both the boxes of human and object targets and the categories of interactions. The overall loss function consists of four components, namely box regression loss $\mathcal{L}_b$, the intersection-over-union loss $\mathcal{L}_u$, the classification loss $\mathcal{L}_c$ and the mimic loss $\mathcal{L}_m$:

$$\mathcal{L} = \lambda_b \sum_{i \in (h,o)} \mathcal{L}_b^i + \lambda_u \sum_{j \in (h,o)} \mathcal{L}_u^j + \sum_{k \in (o,a)} \lambda_c^k \mathcal{L}_c^k + \lambda_m \mathcal{L}_m \tag{9}$$

where $\mathcal{L}_m$ represents an auxiliary loss employed for knowledge distillation from the language model to the visual HOI part. This approach has been proven effective for cross-modal learning in the [30], so we retain this design in our work. We follow the GEN-VLKT [30] and QPIC [40] to set the $\lambda_b$, $\lambda_u$, $\lambda_c$ and $\lambda_m$ to 2.5, 1, 1 and 20.

## F  Open-category Inference

In UniHOI, we harness the cross-modal alignment capabilities of the VL foundation model to perform vision-text content alignment. More specifically, we utilize the Image Encoder in BLIP-2 to extract image tokens, and then employ Q-Former to convert Image tokens into interaction representations that can be aligned with the features in text Encoder. The LLM is used to extract the descriptive text of the interaction relationship that needs to be recognized, for which we use the $[CLS]$ token as the text modality representation $\mathcal{T}^{cls}$. Subsequently, the cosine similarity $Sim(\cdot)$ is calculated between the visual modality representation and the text modality representation to determine whether the interaction described in the input text has occurred in the $j$-th HO pair:

$$Logits_j^{hoi} = Sim\left[\mathcal{T}^{cls}, FC(Concat(\mathcal{V}_j^i, \mathcal{V}_j^f))\right] \tag{10}$$

## G  Computational Overhead

Table 9 showcases the computational overhead of UniHOI-s and GEN-VLKT methods of varying sizes when processing an $800 \times 600$ image, as well as their test results on the HICO-DET dataset. Here, GEN-VLKT-s, GEN-VLKT-m, and GEN-VLKT-l are structures proposed in [30]. To further investigate the HOI recognition performance of larger models, we extended the number of layers in the GEN-VLKT interaction decoder to 10 and 20 layers respectively, thus designing two variants, GEN-VLKT-xl and GEN-VLKT-xxl. Our results on the HICO-DET dataset suggest that continuing to enlarge the model following the original technical route of the visual HOI detector only yields limited improvement in HOI recognition performance. In comparison, our proposed UniHOI displays a significant accuracy advantage over larger visual HOI detectors, corroborating the superiority of our framework.

Table 9: Test results of different methods on HICO-DET test set.

| Method | Inference Time (ms) | Default Setting | | | Known Objects Setting | | |
| --- | --- | --- | --- | --- | --- | --- | --- |
| | | Full | Rare | Non-rare | Full | Rare | Non-rare |
| GEN-VLKT-s | 30.31 | 33.75 | 29.25 | 35.10 | 36.78 | 32.75 | 37.99 |
| GEN-VLKT-m | 34.92 | 34.78 | 31.50 | 35.77 | 38.07 | 34.94 | 39.01 |
| GEN-VLKT-l | 45.62 | 34.96 | 31.18 | 36.08 | 38.22 | 34.36 | 39.37 |
| GEN-VLKT-xl | 58.96 | 35.03 | 31.27 | 35.97 | 38.17 | 34.41 | 39.45 |
| GEN-VLKT-xxl | 91.57 | 35.23 | 31.42 | 36.17 | 38.39 | 34.62 | 39.68 |
| *UniHOI-s* (Ours) | 85.86 | 40.06 | 39.91 | 40.11 | 42.20 | 42.60 | 42.08 |
| | | (+6.31) | (+10.66) | (+5.01) | (+5.42) | (+9.85) | (+4.09) |

## H  Datasets and Evaluation Metrics

**V-COCO** [15]. V-COCO is a specialized derivative of the MS-COCO dataset, composed of 5,400 images allocated for training and 4,946 set aside for testing. This dataset encompasses a broad range

of categories, specifically 80 object categories, 29 interaction categories, and an extensive total of 234 HOI categories. In adherence with previously established methodologies [27], our evaluation focuses on a subset of 24 interaction classes. This choice is justified by the fact that four interaction classes lack object pairings, and one class suffers from a paucity of samples. For our evaluation metrics, we rely on the Mean Average Precision (mAP). For scenarios involving object occlusion, two different evaluation approaches are entertained. Scenario 1 adopts a stringent evaluation criterion necessitating a null bounding box prediction with coordinates [0, 0, 0, 0], while Scenario 2 provides a more lenient condition, disregarding the predicted bounding box in such occlusion instances for the purpose of evaluation.

**HICO-DET** [7]. HICO-DET is an expansive dataset designed for HOI detection, consisting of 37,536 images for training and an additional 9,515 for testing. It includes a considerable spectrum of categories, namely, 80 object categories, 117 interaction categories, and a substantial total of 600 HOI categories. Our evaluation approach aligns with prior methodologies [27], focusing on the HICO-DET. We calculate the mAP metric under two distinct settings: Default and Known Objects. These settings are applied across three categories: Full (spanning all 600 HOI classes), Rare (consisting of 138 classes with fewer than 10 training samples), and Non-rare (comprising 462 classes with more than 10 training samples).