# OpenReview forum: "Detecting Any Human-Object Interaction Relationship: Universal HOI Detector with Spatial Prompt Learning on Foundation Models"
_NeurIPS.cc/2023/Conference — NeurIPS 2023 poster_

### Official Review · Reviewer_X6bU · 2023-07-04

**Soundness:** 3 good
**Presentation:** 2 fair
**Contribution:** 2 fair
**Rating:** 6
**Confidence:** 4

**Summary:**

This paper proposes a framework for human-object interaction (HOI) detection that leverages vision-language foundation models and large language models to achieve universal and flexible recognition of complex interactions in images. The framework, named UniHOI, consists of three main components: a visual HOI detector that extracts three levels of features from images, a HO prompt-guided decoder that queries the foundation model for high-level relation representations associated with human-object pairs, and a knowledge retrieval module that uses a large language model to generate descriptive texts for interaction categories. The framework supports both supervised and zero-shot settings, and can handle any textual input for open-category interaction detection. The paper demonstrates the effectiveness and superiority of UniHOI over existing methods on two public benchmarks, HICO-DET and V-COCO, as well as in the wild scenarios.

**Strengths:**

- The performance is quite impressive.
- Leveraging LLM and foundational models to augment CV tasks is the future, and this work gives a try to use them simultaneously.
- The authors promise to release the code to ensure reproducibility.

**Weaknesses:**

1. There is no ablative study of each component (perhaps only one component, i.e., HO prompt decoder) under the close-set setup.
2. This work uses BLIP2 with ViT-L while existing work like GEN-VLKT typically uses CLIP with ViT-B. It is evident that the former is much more powerful. Could you provide the performance on HICO-DET using HO prompt-based learning with CLIP ViT-B under the close-set setup? As the improvement may be brought by more advanced large visual-language pre-trained models.
3. How about the inference speed? As shown in Table 8, **three** times longer than GEN-VLKT. Note that existing work like GEN-VLKT does not involve the computation of large visual-language pre-trained models at the inference stage, since all of the feature of objects or verbs is pre-computed. However, in this work, the feature for prompting should be computed for each image individually. Considering the extremely large backbone (e.g, ViT-L), there would be a heavy burden at inference.
4. The core contribution of this work is actually the HO prompt-based decoder. However, there is nothing novel with it, i.e., directly using spatial location to get foundational model output features.
5. Knowledge retrieval is solely used in open-world setup, is it possible for it to augment the closed-world setup?

Overall, this is technically solid work, and LLM for knowledge retrieval is interesting. But the comparison is unfair (i.e., a much more powerful visual-language pre-trained model), the inference time is unacceptable, and the novelty of the prompt-based decoder is limited. I will be very happy to update my score if the authors can address my concerns above.

**Questions:**

Please refer to weaknesses.

**Limitations:**

There is no discussion on limitations or failure cases.

---

> ### Author Rebuttal · Authors · 2023-08-10
>
> Dear Reviewer X6bU,
>
> First and foremost, we'd like to express our gratitude for your comprehensive review and insightful comments. We acknowledge the concerns you've raised and will attempt to address them point by point:
>
>
> **Ablations**: We appreciate the importance of ablation studies to establish the individual contributions of each component. We would like to kindly draw your attention to Table 5, which presents our ablation experiments conducted under the closed-set setup, and to Table 6, which depicts results from the open-set setup.
>
>
> **Model Comparison with GEN-VLKT using CLIP ViT-B**: Thank you for the keen observation regarding the comparative power of BLIP2 with ViT-L and CLIP with ViT-B. We wholeheartedly agree on the importance of a fair comparison. In line with this, we have conducted experiments using CLIP ViT-B not just in the close-set setup, but also in the open-set setup. We have conducted experiments on V-COCO datasets:
>
> |Method|${AP}^{1}_{role}$|${AP}^{2}_{role}$|
> |:-:|:-:|:-:|
> |GEN-VLKT_s|62.41|64.46|
> |${UniHOI}_s$ (w/ CLIP)|63.79|65.91|
> |${UniHOI}_s$ (w/ BLIP2)|65.58|68.27|
> |GEN-VLKT_m|63.28|65.58|
> |${UniHOI}_m$ (w/ CLIP)|64.47|67.83|
> |${UniHOI}_m$ (w/ BLIP2)|67.95|70.61|
> |GEN-VLKT_l|63.58|65.93|
> |${UniHOI}_l$ (w/ CLIP)|64.86|67.98|
> |${UniHOI}_l$ (w/ BLIP2)|68.05|70.82|
>
> The following table shows the results of UniHOI equipped with different foundation methods on the HICO-DET dataset:
>
> |||Default|||Known Obj.||
> |:-:|:-:|:-:|:-:|:-:|:-:|:-:|
> |Method|Full|Rare|Non-rare|Full|Rare|Non-rare|
> |GEN-VLKT_s|33.75|29.25|35.10|36.78|32.75|37.99|
> |${UniHOI}_s$ (w/ CLIP)|35.92|34.39|36.26|38.84|37.19|40.18|
> |${UniHOI}_s$ (w/ BLIP2)|42.73|42.03|42.93|43.58|45.27|43.08|
> |GEN-VLKT_m|34.78|31.50|35.77|38.07|34.94|39.01|
> |${UniHOI}_m$ (w/ CLIP)|36.71|35.42|36.91|39.16|39.23|40.56|
> |${UniHOI}_m$ (w/ BLIP2)|43.18|42.98|43.57|44.32|45.96|43.89|
> |GEN-VLKT_l|34.96|31.18|36.08|38.22|34.36|39.37|
> |${UniHOI}_l$ (w/ CLIP)|36.84|35.71|37.05|39.28|39.31|40.79|
> |${UniHOI}_l$ (w/ BLIP2)|43.57|43.79|44.01|44.78|46.48|44.32|
>
> Additionally, the results under zero-shot setting are as follows:
>
> |Method|Type|Unseen|Seen|Full|
> |:-:|:-:|:-:|:-:|:-:|
> |GEN-VLKT_s|RF-UC|21.36|32.91|30.56|
> |${UniHOI}_s$ (w/ CLIP)|RF-UC|23.41|33.45|31.97|
> |${UniHOI}_s$ (w/ BLIP2)|RF-UC|28.68|33.16|32.27|
> |GEN-VLKT_s|NF-UC|25.05|23.38|23.71|
> |${UniHOI}_s$ (w/ CLIP)|NF-UC|26.89|25.57|25.96|
> |${UniHOI}_s$ (w/ BLIP2)|NF-UC|28.45|32.63|31.79|
> |GEN-VLKT_s|UO|10.51|28.92|25.63|
> |${UniHOI}_s$ (w/ CLIP)|UO|13.24|30.27|27.52|
> |${UniHOI}_s$ (w/ BLIP2)|UO|19.72|34.76|31.56|
> |GEN-VLKT_s|UV|20.96|30.23|28.74|
> |${UniHOI}_s$ (w/ CLIP)|UV|22.18|33.29|30.87|
> |${UniHOI}_s$ (w/ BLIP2)|UV|26.05|36.78|34.68|
>
>
> Our experiments demonstrate that **regardless of whether we use CLIP or BLIP2, our approach significantly outperforms the current state-of-the-art methods**.
>
> In addition, one of the motivations we chose for BLIP is that **the training text for CLIP is too simple**, such as "an image of an apple". **We are concerned that the knowledge in CLIP may not be sufficient for challenging HOI detection**. In order to explore the potential of the latest large models, the default UniHOI has chosen BLIP2, which is pre trained with richer text.
>
> **Inference Speed**: Thank you for highlighting the computational aspects of UniHOI. In response:
>
> - We've mitigated computational demands by adjusting the image resolution for the VL foundation model, striking a balance between efficiency and performance.
>
> - For practical applications, we're exploring techniques like quantization to optimize inference speed without sacrificing results.
>
> - Addressing the challenges posed by large models remains an ongoing effort, with potential solutions in architectural refinements or improved knowledge transfer methods.
>
>
> **Novelty of HO Prompt-based Decoder**: Thank you for drawing attention to the novelty. While the core concept may appear direct, the effectiveness lies in its integration and synergy with other components. Here's a deeper dive into the innovative aspects of our approach:
>
> - UniHOI is the pioneering approach using prompt learning for VL foundation models in HOI detection, surpassing state-of-the-art methods in both supervised and zero-shot settings. This sets a more reliable benchmark for harmonizing specialized HOI detectors with large models.
>
> - UniHOI efficiently extracts spatial representations of humans and objects from professional HOI detectors. This enables us to conveniently and accurately capture instance-level features from large models. Previous approaches (e.g., GEN-VLKT, HOICLIP) driven by large models predominantly aligned only with image-level features, overlooking this critical facet.
>
> - UniHOI unleashes the potential of VL foundation models and LLMs (e.g., GPT-4), potentially inspiring future endeavors to explore more universal and intelligent HOI detectors.
>
> - The elegance of our core concept is in its simplicity and directness, making it adaptable across various VL foundation models.
>
>
> **Knowledge Retrieval in Closed-World Setup**: Thank you for the insightful suggestion. In response, we have added experiments on the Knowledge Retrieval (KR) in the closed-world setup. The results are as follows:
>
> |Method|${AP}^{1}_{role}$|${AP}^{2}_{role}$|
> |:-:|:-:|:-:|
> |GEN-VLKT_s|62.41|64.46|
> |${UniHOI}_s$|65.58|68.27|
> |${UniHOI}_s$+KR|66.12|68.73|
> |GEN-VLKT_m|63.28|65.58|
> |${UniHOI}_m$|67.95|70.61|
> |${UniHOI}_m$+KR|68.59|71.13|
> |GEN-VLKT_l|63.47|65.93|
> |${UniHOI}_l$|68.05|70.82|
> |${UniHOI}_l$+KR|69.39|71.24|
>
> The results indicate that **knowledge retrieval can also promote model performance in Closed-World scenarios**. This is because more detailed textual description information can promote a deeper understanding of interactions.
>
> We genuinely appreciate your time and expertise. If you have any further questions, please let us know. We’d be very happy to do anything we can that would be helpful in the time remaining! Thanks!

---

> > ### Author Response · Authors · 2023-08-14
> > **Further Discussion with Reviewer X6bU**
> >
> > Dear Reviewer X6bU,
> >
> > We sincerely appreciate the time you devoted to reviewing our manuscript and the invaluable feedback you provided. We have diligently addressed your comments and provided corresponding responses and results. We believe that these responses adequately address the concerns you raised. We would be grateful for an opportunity to discuss whether your reservations have been resolved. Should there be any aspect of our work that remains unclear, please do not hesitate to inform us.
> >
> > Once again, thank you for your constructive insights.
> >
> > Warm regards,

---

> > > ### Comment · Reviewer_X6bU · 2023-08-20
> > >
> > > Thank you for the response. The rebuttal addresses most of my concerns and I will raise my score to 6. Please incorporate the revision into the final version.

---

### Official Review · Reviewer_djfM · 2023-07-04

**Soundness:** 3 good
**Presentation:** 2 fair
**Contribution:** 3 good
**Rating:** 5
**Confidence:** 5

**Summary:**

In view of the limited scalability and the suboptimal zero-shot performance of current HOI detection methods, the authors propose a novel method for HOI detection based on VL foundation models.
With in-depth analysis and adaptation of HOI detectors, the foundation model is effectively adopted to reason about HOI relationships based on human/object tokens.
Furthermore, LLM is adopted as a knowledge base to diversify HOI descriptions, enabling open-vocabulary HOI detections.
With the VL foundation model and LLM, extraordinary HOI detection performance is achieved upon both conventional and zero-shot setting.

**Strengths:**

The proposed HO prompt-guided decoder is brilliant in addressing the feature alignment issue.

Adopting GPT as knowledge base is an interesting idea to incorporate the recent progress in LLM with HOI detection.

The performance is amazing with significant margins upon previous SOTAs, especially for zero-shot setting.

Extensive experiments are conducted, providing valuable insights on the effect of VL foundation models in HOI detection.

In the wild HOI detection illustration is quite impressive.


**Weaknesses:**

The comparison between GEN-VLKT and the proposed method is not totally fair to me. It might be better to replace BLIP-2 with CLIP for a fair comparison.

Fig. 2 is not very clear. It would help if annotating the encoders in the figure with corresponding notations.

The baseline of ablation experiments is chosen as GEN-VLKT. However, a major difference between GEN-VLKT and UniHOI is the VL foundation model used. And there are also other differences, like VLKT is not used for UniHOI. It might be better to change the baseline to make the ablation more reasonable.

**Questions:**

- The description on the adopted Image Encoder and Instance Decoder is not clear. Is DETR adopted? Or some more advanced detectors? Are they frozen during training?

- In L177, $V^f$ and $V^f$ seems to be a typo.

- The performance of UniHOI-l in Tab. 1-2 is questionable. Please check whether there are typos.

- In the ablation studies, it is still not very clear that how the VL Foundation model is simply added. Is it a simple removal of HO Spatial Prompting? Or replacing the input feature of HO Spatial Prompting with the learnable queries?

- Is it possible to make the HOI detector share the backbone of VL foundation model? This could be related to the proposed insight that HOI detector is a three-tier visual feature hierarchy. Ablation studies on this would be preferrable.

- The proposed method seems to be applicable to arbitrary HOI detectors (if the answer to Q1 is yes). Is it practical?

- Results in Tab. 5 and Tab. 3 are not consistent. The result without Knowledge Retrieval is reported in Tab. 3.

**Limitations:**

Limitations of the paper is not well-discussed in the paper.
I would like to see more discussion on extending the use of LLM further than a static knowledge base.
Also please refer to the weakness part.

---

> ### Author Rebuttal · Authors · 2023-08-10
>
> Dear Reviewer djfM,
>
> We truly appreciate the detailed feedback. Herein, we provide a detailed response to each of your concerns:
>
> **On Weaknesses**:
>
> a. **Comparison with GEN-VLKT**: In response, we replace BLIP2 with CLIP and conducted experiments on V-COCO datasets:
>
> |Method|${AP}^{1}_{role}$|${AP}^{2}_{role}$|
> |:-:|:-:|:-:|
> |GEN-VLKT_s|62.41|64.46|
> |${UniHOI}_s$ (w/ CLIP)|63.79|65.91|
> |${UniHOI}_s$ (w/ BLIP2)|65.58|68.27|
> |GEN-VLKT_m|63.28|65.58|
> |${UniHOI}_m$ (w/ CLIP)|64.47|67.83|
> |${UniHOI}_m$ (w/ BLIP2)|67.95|70.61|
> |GEN-VLKT_l|63.58|65.93|
> |${UniHOI}_l$ (w/ CLIP)|64.86|67.98|
> |${UniHOI}_l$ (w/ BLIP2)|68.05|70.82|
>
> The following table shows the results of UniHOI equipped with different foundation methods on the HICO-DET dataset:
>
> |||Default|||Known Obj.||
> |:-:|:-:|:-:|:-:|:-:|:-:|:-:|
> |Method|Full|Rare|Non-rare|Full|Rare|Non-rare|
> |GEN-VLKT_s|33.75|29.25|35.10|36.78|32.75|37.99|
> |${UniHOI}_s$ (w/ CLIP)|35.92|34.39|36.26|38.84|37.19|40.18|
> |${UniHOI}_s$ (w/ BLIP2)|42.73|42.03|42.93|43.58|45.27|43.08|
> |GEN-VLKT_m|34.78|31.50|35.77|38.07|34.94|39.01|
> |${UniHOI}_m$ (w/ CLIP)|36.71|35.42|36.91|39.16|39.23|40.56|
> |${UniHOI}_m$ (w/ BLIP2)|43.18|42.98|43.57|44.32|45.96|43.89|
> |GEN-VLKT_l|34.96|31.18|36.08|38.22|34.36|39.37|
> |${UniHOI}_l$ (w/ CLIP)|36.84|35.71|37.05|39.28|39.31|40.79|
> |${UniHOI}_l$ (w/ BLIP2)|43.57|43.79|44.01|44.78|46.48|44.32|
>
> Additionally, the results under zero-shot setting are as follows:
>
> |Method|Type|Unseen|Seen|Full|
> |:-:|:-:|:-:|:-:|:-:|
> |GEN-VLKT_s|RF-UC|21.36|32.91|30.56|
> |${UniHOI}_s$ (w/ CLIP)|RF-UC|23.41|33.45|31.97|
> |${UniHOI}_s$ (w/ BLIP2)|RF-UC|28.68|33.16|32.27|
> |GEN-VLKT_s|NF-UC|25.05|23.38|23.71|
> |${UniHOI}_s$ (w/ CLIP)|NF-UC|26.89|25.57|25.96|
> |${UniHOI}_s$ (w/ BLIP2)|NF-UC|28.45|32.63|31.79|
> |GEN-VLKT_s|UO|10.51|28.92|25.63|
> |${UniHOI}_s$ (w/ CLIP)|UO|13.24|30.27|27.52|
> |${UniHOI}_s$ (w/ BLIP2)|UO|19.72|34.76|31.56|
> |GEN-VLKT_s|UV|20.96|30.23|28.74|
> |${UniHOI}_s$ (w/ CLIP)|UV|22.18|33.29|30.87|
> |${UniHOI}_s$ (w/ BLIP2)|UV|26.05|36.78|34.68|
>
> These results demonstrate that **regardless of whether we use CLIP or BLIP2, our approach significantly outperforms the current state-of-the-art methods**.
>
> b. **Clarification on Fig. 2**: We will add annotations to all the components in Fig. 2 to ensure clarity.
>
> c. **Baseline in Ablation Experiments**: Based on your suggestion, we adopted a GEN that doesn't use CLIP model as our baseline and performed ablation experiments by equipping BLIP2. The results are as follows:
> |Method|${AP}^{1}_{role}$|${AP}^{2}_{role}$|
> |:-:|:-:|:-:|
> |baseline|61.58|63.59|
> |+ BLIP2|62.91|64.83|
> |+ HOPD|65.58|68.27|
> |+ Knowledge Retrieval|66.74|69.31|
>
>  Furthermore, we also conduct ablation experiments using CLIP as the VL foundation model:
>
> |Method|${AP}^{1}_{role}$|${AP}^{2}_{role}$|
> |:-:|:-:|:-:|
> |baseline|61.58|63.59|
> |+ CLIP|62.15|64.28|
> |+ HOPD|63.79|65.91|
> |+ Knowledge Retrieval|63.98|66.34|
>
> **On Questions**:
>
> a. **Image Encoder & Instance Decoder**: We employed the image encoder and instance decoder from the single-stage HOI detector GEN, allowing their weights to adapt during training, while the weights of the VL foundation model remain frozen.
>
> b. **Typos in L177**: We deeply regret the oversight. These errors will be addressed in the revised manuscript.
>
> c. **Performance of UniHOI-l in Tab. 1-2**: In Table 1, UniHOI-l reports a mAP of 43.79 in the Rare category under the Default Setting, showing a "12.61" improvement over GEN-VLKT-l. However, there was an oversight where we mistakenly wrote "2.61". We will correct this mistake.
>
> d. **Clarification on Ablation Studies**: We use the BLIP2 to produce a representation $X^q$ of dimension [32,768]. Then, a two-layer Transformer Decoder processes a learnable query [64,256] to derive the feature $V^q$ from $X^q$. This is then combined with the output $V^i$ from the Interaction Decoder for our prediction. However, without UniHOI's spatial feature prompting, the feature alignment between $V^q$ and $V^i$ is suboptimal.
>
> e. **Sharing Backbone of VL Foundation Model**: Thanks for the thoughtful suggestion. We employ its Image Encoder to derive image features, yielding a representation of [1024,1408]. We linearly interpolate the positional encodings in BLIP2 to handle larger images and used the Instance Decoder with learnable Queries of [64,256] to identify interacting HO pairs. **Unfortunately, achieving model convergence proved challenging.** Two primary constraints were observed:
>
> - BLIP2's training lacks emphasis on individual localization, limiting its precision in Instance Detection.
>
> - BLIP2's size necessitates a smaller input image. Enlarging this to match typical detection models would produce numerous tokens, adding substantial computational burden during LLM alignment.
>
> f. **Applicability to Arbitrary HOI Detectors**: Our UniHOI is universally adaptable. Quality spatial information from any detector can prompt the VL model to extract advanced features, enhancing versatility across multiple HOI detectors and domains. In the future, we will try to release open source code that is compatible with classic HOI detectors, such as UPT (Unary–Pairwise Transformer).
>
> g. **Consistency in Tab. 5 and Tab. 3**:
>
> - In Table 3, we solely showcased the results without the utilization of knowledge retrieval, which has already achieved substantial performance enhancements.
>
> - In this paper, we treated knowledge retrieval as an ancillary mechanism to elevate the model's performance, specifically aiding in comprehending intricate interactions.
>
> - We commit to sharing the relevant code, weights, and results to support further exploration of knowledge-based techniques.
>
> **On Limitations**: We plan to enhance our conclusion, addressing potential areas such as model compression, knowledge representation, parameter-efficient tuning, and multi-turn dialogues.
>
> We genuinely appreciate your time and expertise. If you have any further questions or concerns, please let us know. Thanks!

---

> > ### Author Response · Authors · 2023-08-14
> > **Further Discussion with Reviewer djfM**
> >
> > Dear Reviewer djfM,
> >
> > Thank you for the time and effort you dedicated to reviewing our manuscript. We sincerely appreciate your valuable feedback. In response to your comments, we have provided thorough explanations and updated results. We believe that these address the concerns you raised. We are eager to ensure that all of your concerns have been addressed adequately. Should there be any aspect of our work that remains unclear to you, please do not hesitate to inform us.
> >
> > Once again, we extend our gratitude for your constructive feedback.
> >
> > Best,

---

### Official Review · Reviewer_WF8N · 2023-07-04

**Soundness:** 4 excellent
**Presentation:** 3 good
**Contribution:** 3 good
**Rating:** 7
**Confidence:** 5

**Summary:**

This paper investigates the problem of human-object interaction (HOI) detection. The authors introduced UniHOI, a method for universal HOI detection in an open-world setting. They also explored the universal interaction recognition with Vision-Language (VL) foundation models and large language models (LLMs), and proposed HO prompt-based learning for high-level relation extraction aimed at VL foundation models.  Experimental results show the effectiveness and significance of the proposed method.

**Strengths:**

1. Overall, the manuscript is well-written and easy to follow. The figures are pretty and can convey the concepts clearly.
2. Pushing the problem of human-object interaction detection toward an open-world setting is of great importance. This is also a trend for most existing computer vision applications.
3. The extensive experimental results show the superiority of the proposed UniHOI method.

**Weaknesses:**

1. In Table 4, the results from the third row come from "ConsNet [31]" but not "ATL [15]", according to the paper of "GEN-VLKT".
2. The conclusion part lacks objective reflections on the deficiencies of this study and future prospects for improvements.

**Questions:**

Please refer to the weakness section.

**Limitations:**

Please refer to the weakness section.

---

> ### Author Rebuttal · Authors · 2023-08-10
>
>
> Dear Reviewer WF8N,
>
> We greatly appreciate your thoughtful review and the time you have taken to provide insights and feedback on our submission. We are encouraged by the positive aspects you've highlighted and grateful for the critical points you've raised. Here, we address the weaknesses mentioned and hope to provide clarifications that would enhance the clarity of our contribution.
>
> **Reference to Table 4 Results**
> We apologize for the oversight. You are right. The result in the third row of Table 4 should reference "ConsNet [31]" and not "ATL [15]". We have duly noted this mistake and will rectify it in the final version of our manuscript. We understand that accuracy in presenting prior works is paramount, and we thank you for pointing out this error.
>
> **Reflections on the Study's Deficiencies and Future Directions**
> Thank you for your constructive feedback. In light of your feedback, we will expand our conclusion section to incorporate the following points:
>
> 1. **Deficiencies**: We acknowledge that while our UniHOI method has shown promise in various experimental settings, it also faces the issue of ***substantial computational demands*** when driving large models.
> 2. **Future Prospects**: Building on our current findings, there are several promising directions: (1) At present, the focus of our UniHOI is mainly on ***Prompt Learning***. In the future, we will consider paradigms such as ***Adapter*** and ***Parameter-Efficient Fine-Tuning*** to further enhance the generic Visual-Language (VL) foundation models' capabilities in the specialized Human-Object Interaction (HOI) domain. (2) We will explore better ***knowledge transfer*** methods in the future to more efficiently implement the driving or collaboration of specialized HOI detectors by VL base models, thereby reducing the inference costs of large models.
>
> In summary, we sincerely thank you for your valuable comments. We believe that by addressing these points, our work will be significantly improved and provide a solid contribution to the community. We hope our responses provide clarity, and we remain open to further feedback. Thanks!

---

> > ### Comment · Reviewer_WF8N · 2023-08-11
> >
> > Thanks for the detailed feedback from the authors. I look forward to seeing some PEFT methods applied to the HOI domain in the near future.

---

### Official Review · Reviewer_pXLy · 2023-07-06

**Soundness:** 3 good
**Presentation:** 3 good
**Contribution:** 2 fair
**Rating:** 4
**Confidence:** 5

**Summary:**

The paper addresses human-object interaction (HOI) detection task. The authors propose a new method named as UniHOI, achieved by prompting BLIP2 using human-object paired features, as well as linguistic semantics generated by a LLM. The proposed UniHOI demonstrates significant performance gain on HICO-DET and V-COCO in both fullly supervised and zero-shot scenarios.

**Strengths:**

If the BLIP2 is a fundamental model pre-trained on a lower-level task than HOI detection (e.g., classification, object detection), I would say this paper is an excellent work  in terms of both model design and performance. Actually, I think this is the first work to transfer a fundamental model into the domain of HOI detection, which may open doors to further exploration on prompting learning  for HOI. However, I cannot accept the choice of using BLIP2 as the fundamental model for HOI detection. I will explain the reasons for this in the  weaknesses below.

**Weaknesses:**

1. Prompting engineering aims to transfer a fundamental model pretrained on **lower-level tasks** to **higher-level** tasks. At a minimum, **the task for pre-training needs to be decoupled from the downstream task**. Otherwise, transferring a model pertrained on higher-level tasks to low-level tasks is not prompting, but fine-tuning. BLIP2 is a powerful model pertrained for VQA, image captioning, and similar tasks. However,  as widely acknowledged,  HOI detection is a sub-problem for  these detailed scene understanding tasks.  Namely, BLIP2 itself is a powerful HOI detector (I have tried using BLIP2 directly for HOI detection, and the performance is impressive).  From this point, BLIP2 cannot be used as a fundamental model for HOI detection since it has a great capability for HOI detection by itself, and is capable of even higher-level tasks. Therefore, this paper is more like a work that fine-tunes BLIP2 on HICO-DET and V-COCO, at the cost of giving up the ability to use BLIP2 for other tasks, e.g., captioning.
2. While direct use of BLIP2 for HOI detection may fail to achieve as impressive performance as that of UniHOI on HICO-DET and V-COCO, **BLIP2 has already achieved the goal of doing HOI detection, i.e., detailed scene understanding**.  Therefore, is it a case of putting the cart before the horse to use BLIP2 for HOI detection only?
3. In a real open-world scenario, I think BLIP2 is more capable of HOI detection compared to UniHOI. After all, the  zero-shot HOI detection capability of UniHOI is mainly inherited from BLIP2.
4. The performance of UniHOI on HICO-DET and V-COCO is impressive.  I think this is the first work that achieves a mAP  being larger than 40% on HICO-DET. However, the comparison is not so fair. As aforementioned, BLIP2 itself is a powerful HOI detector, which has been pre-trained with a large amount of **interaction-specific** data.  Note that, **on a fair comparison,  a fundamental model should not be pre-trained using data with annotations involving downstream tasks**. Otherwise, the authors need to report the results without using these extra data. For instance, if we first collect all data involving HOI detection from the dataset used for BLIP2 pre-training. Next, we use these data  to pre-train a HOI detector listed in Table 1, (e.g., GEN-VLKT) to get GEN-VLKT-2. Finally, we fine-tune the GEN-VLKT-2 on the HICO-DET and V-COCO. I think it can also get an excellent performance. This is another reason why I think that transferring a model pre-trained on a higher-level task to a lower- level task (especially when the lower-level task is a sub-task of the higher-level task) is not a prompting, but a fine-tuning.


**Questions:**

Actually, I am very much looking forward to the work of using fundamental models for HOI detection. However, I think an instrive work is to transfer a fundamental model pretrained on lower-level task to a high-level task. This work, however, seems to be the opposite.

---

> ### Author Rebuttal · Authors · 2023-08-10
>
> Dear Reviewer pXLy,
>
> First and foremost, we extend our deepest gratitude for your insightful feedback and hope our clarifications address your concerns. We're eager to highlight the significance and potential of our work.
>
> **1. Regarding BLIP2's Utilization**:
>
> We agree with your statement about Prompting engineering. However, with the utmost respect, I'd like to disagree with the assertion that "utilizing BLIP2 is a fine-tuning process on the HOI task." Please allow me to present several clarifications:
>
> - **Higher-level vs. Lower-level Task Pre-training**: BLIP2's pretraining revolves around global image-level tasks such as Image-Text Contrastive Learning (ITC), Image-grounded Text Generation (ITG), and Image-Text Matching (ITM). On the other hand, HOI detection involves detailed instance-level tasks like **instance localization** and interaction reasoning. Our UniHOI framework enhances BLIP2's **detection capabilities** beyond its original scope.
>
> - **Fine-tuning vs. Prompting**: Fine-tuning inherently modifies model parameters, as referenced in [1][2]. In our case, all parameters of BLIP2 are **frozen**. We leverage spatial features as prompts for BLIP2 without compromising its other functionalities (e.g. captioning).
>
> - **Motivations**: UniHOI presents a **robust synergy** between a specialized HOI model and a generic VL foundation model. With the foundation model retaining its core capabilities, it significantly enhances the performance of more specialized tasks.
>
> - **Using the Same Foundation Model - CLIP**: As additional evidence, we introduced **CLIP-driven UniHOI** to illustrate that our proposed method still achieves remarkable performance on VL foundation models like CLIP, which has been trained on (image, text) pairs for contrastive learning. Prior research like GEN-VLKT and HOICLIP also explored large model-driven methodologies based on CLIP. We've furnished detailed experiments and outcomes regarding this in the subsequent fourth response.
>
> [1] Visual Prompt Tuning, ECCV 2022.
>
> [2] Visual Tuning, arXiv 2023.
>
> **2. BLIP's ability to handle HOI tasks**:
>
> To further explore the possibility of using BLIP2 for HOI detection, we employed its Image Encoder to derive image features, yielding a representation of [1024,1408]. We linearly interpolated the positional encodings in BLIP2 to handle larger images and used the Instance Decoder with learnable Queries of [64,256] to identify interacting HO pairs. **Unfortunately, achieving model convergence proved challenging.** Two primary constraints were observed:
>
> - BLIP2's training lacks emphasis on individual localization, limiting its precision in Instance Detection.
>
> - BLIP2's size necessitates a smaller input image. Enlarging this to match typical detection models would produce numerous tokens, adding substantial computational burden during LLM alignment.
>
> **3. Open-world Scenario**:
>
> Thank you once again for your careful consideration of our work.
>
> - Firstly, we acknowledge that models like CLIP and BLIP2 exhibit superior adaptability in open-world scenarios, which aligns with our experimental findings. We endeavor to capitalize on these models' strengths while tailoring solutions for the nuanced requirements of HOI detection.
>
>
> - Presently, BLIP2's **text-driven interfacer** hinders nuanced interaction analysis. This limitation is evident when trying to describe complex visuals, such as distinguishing individuals in identical jerseys during a sports event, using just text.
>
> - The surge in large models driving specialized tasks, as seen with our UniHOI or the innovative SAM for image segmentation, indicates a promising direction. With UniHOI, we aim to bolster advancements in high-precision HOI detection.
>
>
> **4. Comparison Fairness**:
>
> Building on our earlier discussions, BLIP2's training tasks, namely ITC, ITG, and ITM, are rooted in the "image-text" paradigm, is hardly direct training for GEN-VLKT. For a fair comparison with CLIP-based approaches like GEN-VLKT, we introduced a CLIP-based UniHOI. Below, we present the results from our CLIP-driven UniHOI and GEN-VLKT on the VCOCO dataset:
>
> |Method|${AP}^{1}_{role}$|${AP}^{2}_{role}$|
> |:--:|:--:|:--:|
> |GEN-VLKT_s|62.41|64.46|
> |${UniHOI}_s$ (w/ CLIP)|63.79|65.91|
> |GEN-VLKT_m|63.28|65.58|
> |${UniHOI}_m$ (w/ CLIP)|64.47|67.83|
> |GEN-VLKT_l|63.58|65.93|
> |${UniHOI}_l$ (w/ CLIP)|64.86|67.98|
>
> We also reported the performance of these two methods on the HICO-DET dataset:
>
> |||Default|||Known Obj.||
> |:--:|:--:|:--:|:--:|:--:|:--:|:--:|
> |Method|Full|Rare|Non-rare|Full|Rare|Non-rare|
> |GEN-VLKT_s|33.75|29.25|35.10|36.78|32.75|37.99|
> |${UniHOI}_s$ (w/ CLIP)|35.92|34.39|36.26|38.84|37.19|40.18|
> |GEN-VLKT_m|34.78|31.50|35.77|38.07|34.94|39.01|
> |${UniHOI}_m$ (w/ CLIP)|36.71|35.42|36.91|39.16|39.23|40.56|
> |GEN-VLKT_l|34.96|31.18|36.08|38.22|34.36|39.37|
> |${UniHOI}_l$ (w/ CLIP)|36.84|35.71|37.05|39.28|39.31|40.79|
>
> Moreover, CLIP driven UniHOI is also significantly better than GEN-VLKT in zero shot settings:
>
> |Method|Type|Unseen|Seen|Full|
> |:--:|:--:|:--:|:--:|:--:|
> |GEN-VLKT_s|RF-UC|21.36|32.91|30.56|
> |${UniHOI}_s$ (w/ CLIP)|RF-UC|23.41|33.45|31.97|
> |GEN-VLKT_s|NF-UC|25.05|23.38|23.71|
> |${UniHOI}_s$ (w/ CLIP)|NF-UC|26.89|25.57|25.96|
> |GEN-VLKT_s|UO|10.51|28.92|25.63|
> |${UniHOI}_s$ (w/ CLIP)|UO|13.24|30.27|27.52|
> |GEN-VLKT_s|UV|20.96|30.23|28.74|
> |${UniHOI}_s$ (w/ CLIP)|UV|22.18|33.29|30.87|
>
>
> In our additional experiments utilizing fair CLIP as the foundational model, UniHOI consistently demonstrated impressive results, further attesting to our method's efficacy.
>
>
> Finally, we humbly request you to consider the innovative spirit and potential impact of our work for the broader research community and kindly reconsider our submission. If you have any further questions, please let us know. We’d be very happy to do anything we can that would be helpful in the time remaining! Thanks!

---

> > ### Author Response · Authors · 2023-08-17
> > **Request for Further Discussion**
> >
> > Dear Reviewer pXLy,
> >
> > I hope this message finds you well.
> >
> > Thank you for your thorough and insightful feedback on our submission. We have carefully addressed your comments and supplemented our work with relevant experimental results. If any ambiguity remains, we sincerely invite further inquiries. We genuinely appreciate your time and dedication to reviewing our research. Thanks.
> >
> > Warmest regards,

---

> ### Author Response · Authors · 2023-08-14
> **Further Discussion with Reviewer pXLy**
>
> Dear Reviewer pXLy,
>
> We sincerely appreciate the time you invested in reviewing our submission and your invaluable feedback. We have diligently addressed your comments and provided corresponding responses and results. We believe that these revisions have addressed the concerns you raised. We would be grateful for the opportunity to further discuss whether your concerns have been adequately addressed. If there are any aspects of our work that remain unclear, please do not hesitate to inform us.
>
> Once again, thank you for your guidance and insights.
>
> Warm regards,

---

### Official Review · Reviewer_WxZv · 2023-07-07

**Soundness:** 3 good
**Presentation:** 3 good
**Contribution:** 2 fair
**Rating:** 5
**Confidence:** 4

**Summary:**

This paper proposes a universal HOID pipeline, which utilizes decoded ho-pair feature as spatial prompts to prompt the VL foundation model with the aim to implement effective prompt-based learning on base VL model and extract HOI related features from it. It also proposes knowledge retrieval for HOID in open-category manner by large-scale pretrained language models. Experiments show the effectiveness of this approach in both generic and zero-shot HOID.



**Strengths:**

1. It explores universal interaction recognition by tansferring the rich knowledge inside Vision-Language foundation models and LLMs to HOI pipeline, which broaden the research scope of HOID.
2. The experimental results are promising. In both generic and zero-shot setting, this approach reaches a new state-of-the-art and surpasses previous methods by a substantial margin.


**Weaknesses:**

1.In line152, ‘P_h’ and ‘P_o’ are described as ‘excellent spatial position features’ and further utilized as HO spatial prompts. However, these features are generated by learnable position embedding and query, which is identical as many transformer-based HOID approaches before such as GEN-VLKT. Can you provide some evidence that these feature are indeed ‘excellent’, why could it provide accurate spatial information concerning ho pairs? Or are there some unique designs I overlook?
2.In Line 177, HOPD is designed for the alignment issue between VL foundation model and HOI pipeline. And the output V_f is incorperated with V_i for final predicition of interaction. But the performance of pure V_f, i.e., only utilizing V_f for prediction is unexamined. This results may more directly show the effectiveness of alignment between VL models and HOI pipeline.
3.Some typos. Line 171, ‘the guidance of’ repeated twice. Line 177, ‘V_i’ is mismarked as ‘V_f’.



**Questions:**

Majorly the questions lies in the choice and interpretability of spatial prompts. It’s unclear why these prompts are ‘excellent spatial position features’. Or have you try some experiments on the choice of these prompts?

---

> ### Author Rebuttal · Authors · 2023-08-10
>
> Dear Reviewer WxZv,
>
> First and foremost, we extend our deepest gratitude for your thorough review and insightful feedback. Your recognition of our method is truly appreciated. We concur with your perspective that utilizing $V_f$ exclusively for prediction would offer a more direct testament to the effectiveness of alignment between VL models and the HOI pipeline. We have diligently conducted the pertinent experiments to this effect. Herein, we provide a detailed response to each of your concerns:
>
>
> **1. Regarding the 'excellent spatial position features' of $P_h$ and $P_o$**:
>
> In our study, “$P_h$” and “$P_o$” are two 64*256 features used for predicting Human boxes $B^h$, Object boxes $B^o$, and categories $C^o$. Consequently, “$P_h$” and “$P_o$” are highly correlated with instance-level position and category information. This makes them exemplary spatial position features, perfectly tailored to serve as spatial location representations to prompt the VL foundation models.
>
> We aim to explore the synergy between universal (e.g. BLIP2 and CLIP) and specialized (e.g. GEN-VLKT) models. This innovative design amalgamates traditional HOI detectors, encompassing instance localization and relational reasoning, with VL foundational models. It provides a novel avenue for large model-driven HOI detection, further inspiring the development and implementation of generic algorithms.
>
>
> **2. Regarding the performance of pure $V_f$**:
>
> We are truly grateful for this insightful consideration. **Conducting this experiment does indeed elucidate the effectiveness of alignment between VL models and the HOI pipeline**. We have appended the results of the experiment on V-COCO dataset where only $V_f$ was used for prediction:
>
> |Model|Feature|${AP}^{1}_{role}$|${AP}^{2}_{role}$|
> |:--:|:--:|:--:|:--:|
> ||Only $V_i$|62.41|64.46|
> |${UniHOI}_s$ (w/ BLIP2)|Only $V_f$|64.51|67.26|
> ||$V_i$+$V_f$|**65.58**|**68.27**|
> ||Only $V_i$|63.28|65.58|
> |${UniHOI}_m$ (w/ BLIP2)|Only $V_f$|66.18|68.95|
> ||$V_i$+$V_f$|**67.95**|**70.61**|
> ||Only $V_i$|63.58|65.93|
> |${UniHOI}_l$ (w/ BLIP2)|Only $V_f$|66.25|69.27|
> ||$V_i$+$V_f$|**68.05**|**70.82**|
>
> The experimental results on the V-COCO dataset show that only the features $V_f$ generated by the BLIP2 model have achieved impressive results, second only to the performance of the feature combination of "$V_i $+$V_f $".
>
> Additionally, in light of the concerns raised by reviewer djfM regarding the performance difference between BLIP2 and CLIP, **we have also conducted experiments using UniHOI driven by CLIP**, specifically evaluating its performance when utilizing $V_f$ and $V_i$:
>
> |Model|Feature|${AP}^{1}_{role}$|${AP}^{2}_{role}$|
> |:--:|:--:|:--:|:--:|
> ||Only $V_i$|62.41|64.46|
> |${UniHOI}_s$ (w/ CLIP)|Only $V_f$|62.87|64.91|
> ||$V_i$+$V_f$|**63.79**|**65.91**|
> ||Only $V_i$|63.28|65.58|
> |${UniHOI}_m$ (w/ CLIP)|Only $V_f$|63.74|66.25|
> ||$V_i$+$V_f$|**64.47**|**67.83**|
> ||Only $V_i$|63.58|65.93|
> |${UniHOI}_l$ (w/ CLIP)|Only $V_f$|63.79|66.83|
> ||$V_i$+$V_f$|**64.86**|**67.98**|
>
> In addition to the two close-set experiments mentioned above, **we have also conducted experiments in an open-set setting on the HICO-DET dataset**. The results of **UniHOI equipped with BLIP2** are as follows:
>
> |Type|Feature|Unseen|Seen|Full|
> |:--:|:--:|:--:|:--:|:--:|
> |RF-UC|Only $V_i$|21.36|32.91|30.56|
> |RF-UC|Only $V_f$|28.22|32.18|31.38|
> |RF-UC|$V_i$+$V_f$|**28.68**|**33.16**|**32.27**|
> |NF-UC|Only $V_i$|25.05|23.38|23.71|
> |NF-UC|Only $V_f$|**28.69**|32.51|31.74|
> |NF-UC|$V_i$+$V_f$|28.45|**32.63**|**31.79**|
> |UO|Only $V_i$|10.51|28.92|25.63|
> |UO	|Only $V_f$|**20.91**|**34.88**|**31.78**|
> |UO	|$V_i$+$V_f$|19.72|34.76|31.56|
> |UV|Only $V_i$|20.96|30.23|28.74|
> |UV	|Only $V_f$|**26.81**|36.41|34.51|
> |UV	|$V_i$+$V_f$|26.05|**36.78**|**34.68**|
>
>
> From the results of both our **close-set** and **open-set** experiments, we draw the following conclusions:
>
> - **The alignment between VL models and the HOI pipeline in our UniHOI is particularly effective**. Even when solely leveraging the features $V_f$ from the large models, our approach yields impressive results.
>
> - In the **close-set scenarios**, the feature combination of “$V_i$+$V_f$” emerges as a superior choice, rendering the model to function much like a specialized HOI detector. However, in the **open-set scenarios**, where there's a stronger emphasis on understanding the open world, it's predominantly the $V_f$ feature that plays a pivotal role.
>
> We highly value your feedback and pledge to incorporate these novel experimental data into our final version. This will undoubtedly enhance our paper, facilitating a more comprehensive understanding and evaluation for readers.
>
> **3. Typos**:
>
> We sincerely apologize for the oversight and are grateful for your meticulous attention to detail. Rest assured, these errors will be rectified in the revised manuscript. Furthermore, we commit to thoroughly scrutinizing the entire document to preclude any similar lapses.
>
>
> Thank you again for your constructive feedback. **We will incorporate the corresponding modifications and expansions into the revised paper**. In addition, **the corresponding code and model weights will be open-source to ensure replication**. If you have any further questions, please let us know. We’d be very happy to do anything we can that would be helpful in the time remaining. Thanks!

---

> > ### Author Response · Authors · 2023-08-14
> > **Further Discussion with Reviewer WxZv**
> >
> > Dear Reviewer WxZv,
> >
> > We sincerely appreciate the time and effort you have dedicated to reviewing our submission. We have carefully addressed your comments and provided corresponding responses and results. We believe that these responses and results adequately address your concerns. We would value an opportunity to further discuss whether your reservations have been resolved. Should there remain any aspects of our work that are unclear to you, please do not hesitate to inform us.
> >
> > Once again, thank you for your invaluable feedback.
> >
> > Best,

---

### Decision · Program_Chairs · 2023-09-21

**Decision:**

Accept (poster)

**Comment:**

This paper presents work on human-object interaction detection.  The main contribution of the paper centers around vision-language foundation models in a prompting setup.  The reviewers appreciated the prompt-guided decoder, the leveraging of language data, and impressive empirical results.

In the rebuttal and subsequent discussion, the authors provided additional empirical results and clarifications on the novelty of the method.  Concerns were raised over the perceived use of higher-level tasks (generic language data) for relatively low-level tasks (human-object interactions).  While this is a valid point, the contributions of this paper remain intact.  In particular, the generic nature of the human-object interactions that can be detected using the proposed method, and the impressive empirical results are the reason for the accept decision.  This paper is a substantive contribution to the literature on human-object interaction detection.